# Early NK-cell and T-cell dysfunction marks progression to severe dengue in patients with obesity and healthy weight

Michaela Gregorova[1], Marianna Santopaolo[1,10], Lucy C. Garner [2,10], Rahma F. Hayati [1,10], Divya Diamond[1], Narayan Ramamurthy[2], Vi Thuy Tran[3], Nguyet Minh Nguyen [3], Kate J. Heesom [4], Vuong Lam Nguyen [3,5], Eben Jones[1], Mike Nsubuga [6], Curtis Luscombe[1], Hoa Thi My Vo [3], Chanh Quang Ho[3], Chau Thi Xuan Nguyen[3], Tam Thi Hoai Dong[3], Duyen Thi Le Huynh[3], Tam Thi Cao[7], Andrew D. Davidson [1], Paul Klenerman[2,8], Sophie Yacoub[3,9] & Laura Rivino [1] ✉

Dengue is a mosquito-borne virus infection affecting half of the world's population for which therapies are lacking. The role of T and NK-cells in protection/immunopathogenesis remains unclear for dengue. We performed a longitudinal phenotypic, functional and transcriptional analyses of T and NK-cells in 124 dengue patients using flow cytometry and single-cell RNA-sequencing. We show that T/NK-cell signatures early in infection discriminate patients who develop severe dengue (SD) from those who do not. These signatures are exacerbated in patients with overweight/obesity compared to healthy weight patients, supporting their increased susceptibility to SD. In SD, CD4$^+$/CD8$^+$ T-cells and NK-cells display increased co-inhibitory receptor expression and decreased cytotoxic potential compared to non-SD. Using transcriptional and proteomics approaches we show decreased type-I Interferon responses in SD, suggesting defective innate immunity may underlie NK/T-cell dysfunction. We propose that dysfunctional T and NK-cell signatures underpin dengue pathogenesis and may represent novel targets for immunomodulatory therapy in dengue.

Dengue is caused by dengue virus (DENV), a mosquito-borne orthoflavivirus that infects an estimated 390 million people, causing 300,000 severe dengue (SD) cases and 20,000 deaths yearly in tropical and subtropical countries[1]. Climate change, urbanisation and human mobility are driving a rapid increase in dengue cases[2].

DENV co-circulates as four serotypes (DENV1–4); infection with any DENV serotype can be asymptomatic or cause symptoms ranging from uncomplicated febrile illness to life-threatening SD characterised by increased vascular permeability leading to plasma leakage, potentially hypovolemic shock, organ impairment and haemorrhagic manifestations. The World Health Organization (WHO) classifies dengue cases as

[1]School of Cellular and Molecular Medicine, University of Bristol, Bristol, UK. [2]Translational Gastroenterology and Liver Unit, Nuffield Department of Medicine, University of Oxford, Oxford, UK. [3]Oxford University Clinical Research Unit, Ho Chi Minh City, Vietnam. [4]Bristol Proteomics Facility, School of Biochemistry, University of Bristol, Bristol, UK. [5]University of Medicine and Pharmacy at Ho Chi Minh City, Ho Chi Minh City, Vietnam. [6]Jean Golding Institute, University of Bristol, Bristol, UK. [7]Hospital for Tropical Diseases, Ho Chi Minh City, Vietnam. [8]NIHR Oxford Biomedical Research Centre, Oxford University Hospitals NHS Foundation Trust, Oxford, UK. [9]Centre for Tropical Medicine and Global Health, Oxford University, Oxford, UK. [10]These authors contributed equally: Marianna Santopaolo, Lucy C. Garner, Rahma F. Hayati. ✉e-mail: laura.rivino@bristol.ac.uk

dengue (D), dengue with warning signs (DWS), or severe dengue (SD), the latter two requiring close clinical monitoring[3]. There are no approved therapeutics for dengue, and the two licensed vaccines provide partial protective efficacy[4]. Incomplete understanding of the mechanisms underlying immune protection and progression to SD challenges the development of host-directed therapies and fully protective vaccines. Secondary infection with a different serotype is the most characterised risk factor for SD, with host immunity believed to play a central but poorly understood role in dengue pathogenesis[5]. More recently, obesity has also emerged as an important risk factor[6]. The mechanisms underlying this increased risk remain unclear, but dysfunctional immune responses could be a contributing factor[7]. Alterations of the T-cell response towards pathogens as well as impaired cytotoxic functions of NK-cells during infectious disease and vaccination have been described to occur in adults and children with obesity[8–11].

The contributions of T-cells and NK-cells to protection/immunopathology in dengue are poorly defined. Neutralising antibodies and CD4$^+$/CD8$^+$ T-cells are protective towards DENV infection, however pre-existing cross-reactive immunity may contribute to immunopathology[12–15]. For T-cells, this phenomenon, known as "original antigenic sin" (OAS), postulates that during secondary infection, pre-existing cross-reactive memory T-cells with low affinity for DENV epitopes of the secondary infecting serotype may undergo suboptimal T-cell receptor triggering. This can lead to poor induction of T-cell cytotoxic functions and skewing of cytokine production towards pro-inflammatory cytokines associated with SD[15–18]. However, a study in school children shows that pre-existing tumour necrosis factor (TNF)-α, interferon (IFN)-γ, and interleukin (IL)−2-producing DENV-specific T-cells are protective towards development of a subsequent symptomatic secondary infection, suggesting that the impact of cross-reactive T-cells in dengue is complex and the extent of occurrence of OAS in SD remains poorly understood[19].

NK-cells and T-cells mediate clearance of DENV-infected cells through production of antiviral cytokines such as IFN-γ and secretion of cytotoxic granules containing perforin and granzymes. Studies by us and others show decreased NK-cell expression of CD69, NKp30, granzyme B (GzmB), and perforin in SD compared to non-SD[20,21]. A potential impairment of immune cells mediating viral clearance during SD is consistent with studies showing the association of SD with high/prolonged viraemia and skewing of the NK-cell response from cytotoxic to cytokine-producing[22–24]. It is also consistent with genetic studies showing the association of single-nucleotide polymorphisms (SNPs) in genes involved in CD8$^+$ T/NK-cell cytotoxicity, namely MHC Class I Polypeptide-Related Sequence B (MICB)[25,26], NKG2D and perforin[20], with increased susceptibility to SD.

Here we perform an in-depth phenotypic, functional and transcriptional analyses of NK and T-cell profiles associated with disease outcomes in 124 Vietnamese dengue patients including patients with overweight/obesity (OW/OB) matched by sex, age and illness phase to patients with healthy weight (HW), at two time points of disease (Supplementary Table S1). We show that in SD, CD4$^+$/CD8$^+$ T-cells and NK-cells display increased co-inhibitory receptor expression and decreased cytotoxic potential compared to non-SD, suggesting immune dysfunction of these cells, with some of these features being exacerbated in OW/OB compared to HW patients. We propose that defective type-I IFN signalling, present across multiple cell types in SD patients, may underlie suboptimal NK-cell and T-cell responses in SD patients. Our study provides new insights into the mechanisms underlying the progression to SD in OW/OB and HW dengue patients and paves the way for the evaluation of novel therapeutic avenues for dengue.

## Results
### Distinct T-cell and NK-cell profiles in SD
The kinetics and phenotypic/functional features of T-cell and NK-cell responses were investigated in peripheral blood mononuclear cells

(PBMCs) from patients with SD and non-SD, with each group including patients with HW and OW/OB, at two time points (TP) of disease, admission (TP1, febrile phase, ≤5 days from fever onset) and approx. 3 days later (TP2, post-febrile phase, days 6−9 from fever onset; Fig. 1a). For these analyses, we included a total of 146 PBMC samples from 104 dengue patients (42 patients with matched PBMC samples at TP1 and TP2; 62 patients with PBMCs available for TP2 only). PBMCs were stained with antibodies targeting markers of CD4$^+$/CD8$^+$ T and NK-cell activation/exhaustion (CD38, HLA-DR, PD-1), proliferation (Ki-67), cytotoxicity (CD56, GPR56, GzmB, and perforin), and peripheral tissue homing [cutaneous lymphocyte-associated antigen (CLA), CCR5] and analysed by flow cytometry. In non-SD patients, the frequencies of total NK-cells and NK CD56$^{dim}$ cells, the most abundant NK-cell subset in circulation, are higher at TP1 compared to TP2, in line with the known early activation of NK-cells during acute viral infection; a similar trend is observed in SD HW patients but not in SD OW/OB patients (Fig. 1b; gating strategies shown in Supplementary Fig. S1). These data suggest increasingly altered NK-cell expansion in the SD HW and OW/OB groups. In line with the known kinetics of T-cell activation, the frequencies of CD8$^+$ T-cells increase from TP1 to TP2 consistently across all groups.

We next compared the levels of expression of the analysed markers, assessed as Median Fluorescence Intensity (MFI), within each cell type in patients with matched TP1 and TP2 samples ($N = 41$ patients: $N = 30$ non-SD; $N = 11$ SD). The expression dynamics of most markers follows the same direction (increased/decreased at TP1/TP2) in both severity groups, with changes being more pronounced in non-SD patients, suggesting more dynamic immune changes in this group (Fig. 1c). Analyses of the frequencies of CD4$^+$/CD8$^+$ T-cells and NK-cells expressing the phenotypic/functional markers show increased frequencies at TP1 in SD patients of CD8$^+$ T-cells and NK-cells expressing the skin homing receptor CLA, and decreased frequencies of GPR56$^+$ (cytotoxic) NK-cells (Fig. 1d, e). At TP2, SD patients display decreased frequencies of cytotoxic CD4$^+$/CD8$^+$ T-cells (GzmB$^+$/CD56$^+$) compared to non-SD patients (Fig. 1g, h). From early in infection, PD-1 expression levels are higher in SD versus non-SD in CD4$^+$/CD8$^+$ T and NK-cells, and these remain significantly higher at TP2 for CD8$^+$ T-cells, suggesting prolonged antigenic stimulation of these cells in SD (Fig. 1f, i). PD-1 is a marker of T-cell activation, and prolonged PD-1 expression marks exhausted T-cells following repetitive antigenic stimulation. Linear discriminant analysis (LDA), including all the analysed set of T/NK-cell markers, effectively separates patients based on their disease severity at TP1 (Fig. 1j and Supplementary Fig. S2a). Within the SD group, T/NK-cell profiles are similar in patients who had already progressed to SD at TP1 and those who progressed to SD during their hospitalisation (enrolled at days 2−4 or days 4−5, respectively; Supplementary Fig. S2b), suggesting that these signatures are present early in disease. Analyses of patients stratified by age group show that age does not impact these immune signatures (Supplementary Fig. S2c, d). LDA plots at TP2 also separate patients based on disease severity albeit less strikingly than that at TP1 (Fig. 1k). The LDA analysis of NK/T-cell profiles also discriminates between patients who have normal weight, are overweight or have obesity suggesting an impact of patient BMI on T and NK-cell responses to DENV (Fig. 1l, m). However, T/NK-cell profiles separate patients more effectively based on their dengue disease severity rather than their BMI (Supplementary Fig. S2e). These findings suggest a progressive change of T/NK-cell features, occurring early in infection, in patients with uncomplicated dengue through to patients with SD.

### Increased PD-1$^+$ CD4$^+$ T-cells in SD
We next examined in more detail the phenotypic/functional profiles of CD4$^+$ T-cells at admission and approx. 3 days later, in non-SD and SD patients with HW and OW/OB. CD4$^+$ T-cell activation defined by HLA-

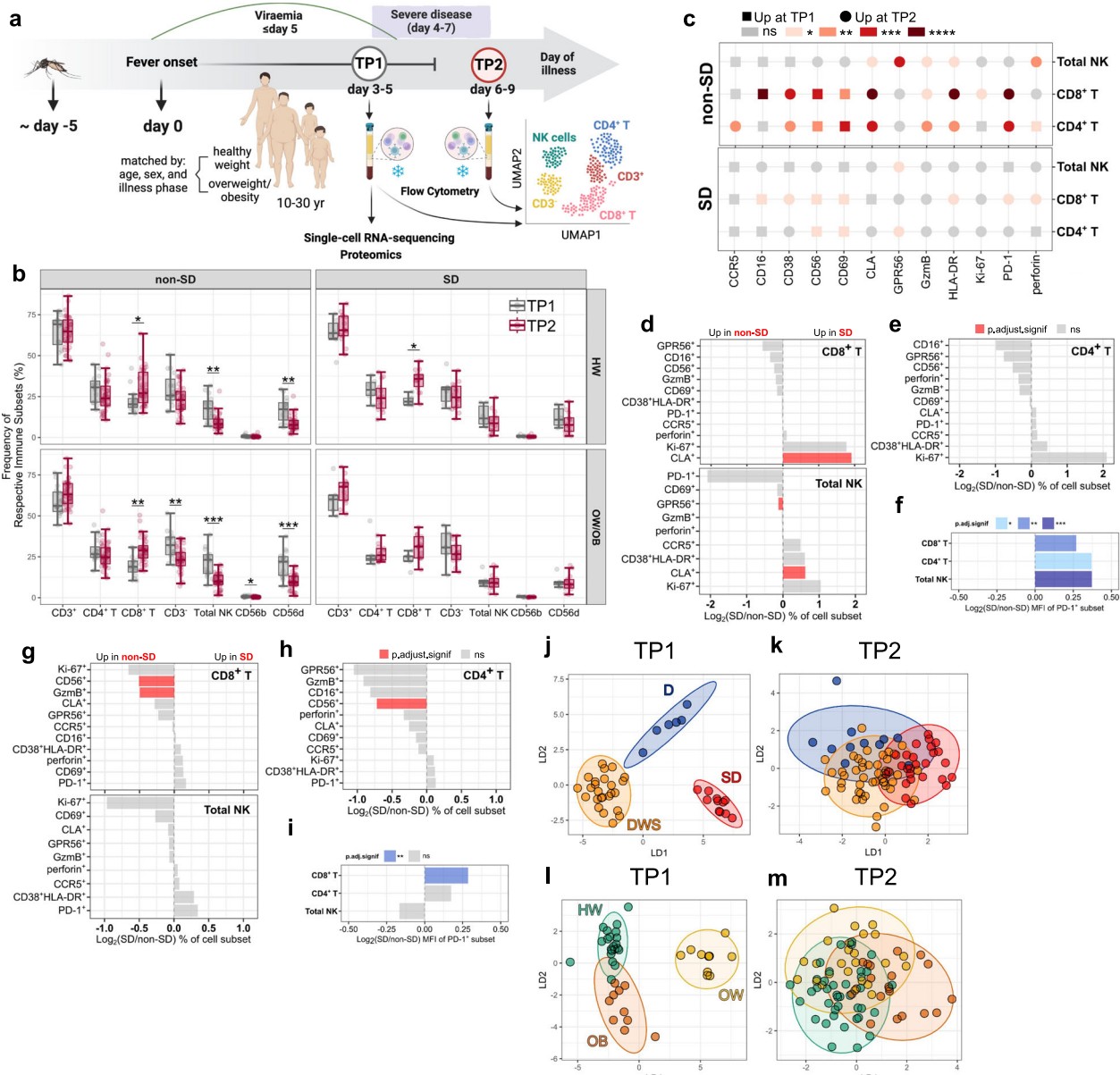

**Fig. 1 | Distinct T-cell and NK-cell profiles in SD. a** Study design showing the two time points (TP) of disease at the indicated time of illness onset (TP1: day 3–5; TP2: day 6–9 shown with a grey and red circle, respectively) and the typical course of a dengue infection with blood viraemia declining around day 5 and severe disease manifestations starting around day 4–5 of illness onset. Created in BioRender. Gregorova, M. (2025) https://BioRender.com/xezlm7c. Flow cytometry analyses of T-cells and NK-cells shown here were performed at TP1 and TP2. **b** Frequency of immune subsets at TP1 [in grey, N = 42: non-SD (N = 31); SD (N = 11); HW (N = 21); OW/OB (N = 21)] and TP2 [in red, N = 104: non-SD (N = 74); SD (N = 27); HW (N = 49); OW/OB (N = 52)]. The middle line in each box represents the median with IQR. **c** Log₂ ratio of Median Fluorescence Intensity (MFI) values of selected markers in total NK, CD8⁺ and CD4⁺ T-cell subsets between TP1 (N = 41) and TP2 (N = 41) in non-SD (N = 30) and SD (N = 11) patients. **d**–**i** Log₂ ratio of mean abundances/PD-1⁺ MFI of

cell subsets between SD and non-SD patients at TP1 (N = 42) (**d**–**f**) and TP2 (N = 104) (**g**–**i**) in CD8⁺ T, CD4⁺ T-cells, and NK-cells. Red/blue bars indicate significance (red bars: *p*.adj. < 0.05; blue bars: \**p*.adj < 0.05; \*\**p*.adj < 0.01; \*\*\**p*.adj < 0.001) calculated using Wilcoxon rank-sum tests. **j**–**m** Linear discriminant analysis (LDA) of T-cell and NK-cell flow cytometry data is shown at TP1 (**j**, **l**; N = 42) and TP2 (**k**, **m**; N = 84). Data points represent individual patients, and symbols/colours indicate disease severity and BMI groups as follows. **j**, **k** Dengue (D), dengue with warning signs (DWS), and severe dengue (SD) are indicated in blue, orange and red circles, respectively. **l**–**m**: Healthy weight (HW), overweight (OW), and obesity (OB) are indicated in green, yellow and orange circles, respectively. Ellipses represent 95% confidence intervals. LD1 and LD2 were derived using all features shown in **d**–**i**. Source data are provided as a Source Data file.

DR/CD38 co-expression increases from TP1 to TP2 and does not differ between non-SD and SD patients (Fig. 2a). PD-1 expression levels also increase from TP1 to TP2, with a trend towards higher expression levels in SD versus non-SD patients at the earlier time point (Fig. 2b). We previously showed that in dengue infection, Ki-67 and CLA co-expression distinguishes DENV-specific T-cells from bystander activated T-cells which lack CLA expression[27,28]. We therefore analysed the features of Ki-67⁺CLA⁺ to gain insights into the features of T-cells

responding to DENV. In non-SD patients, the frequencies of Ki-67⁺CLA⁺ CD4⁺ T-cells increased from TP1 to TP2, mirroring CD4⁺ T-cell activation. However, in SD patients CD4⁺ Ki-67⁺CLA⁺ T-cells are present at higher frequencies early at TP1 compared to non-SD patients, and then fail to expand further, suggesting different kinetics of expansion of responding CD4⁺ T-cells in SD patients (Fig. 2c). The frequencies of Ki-67⁺GzmB⁺ CD4⁺ T-cells follow similar trends as Ki-67⁺CLA⁺ CD4⁺ T-cells (Fig. 2d).

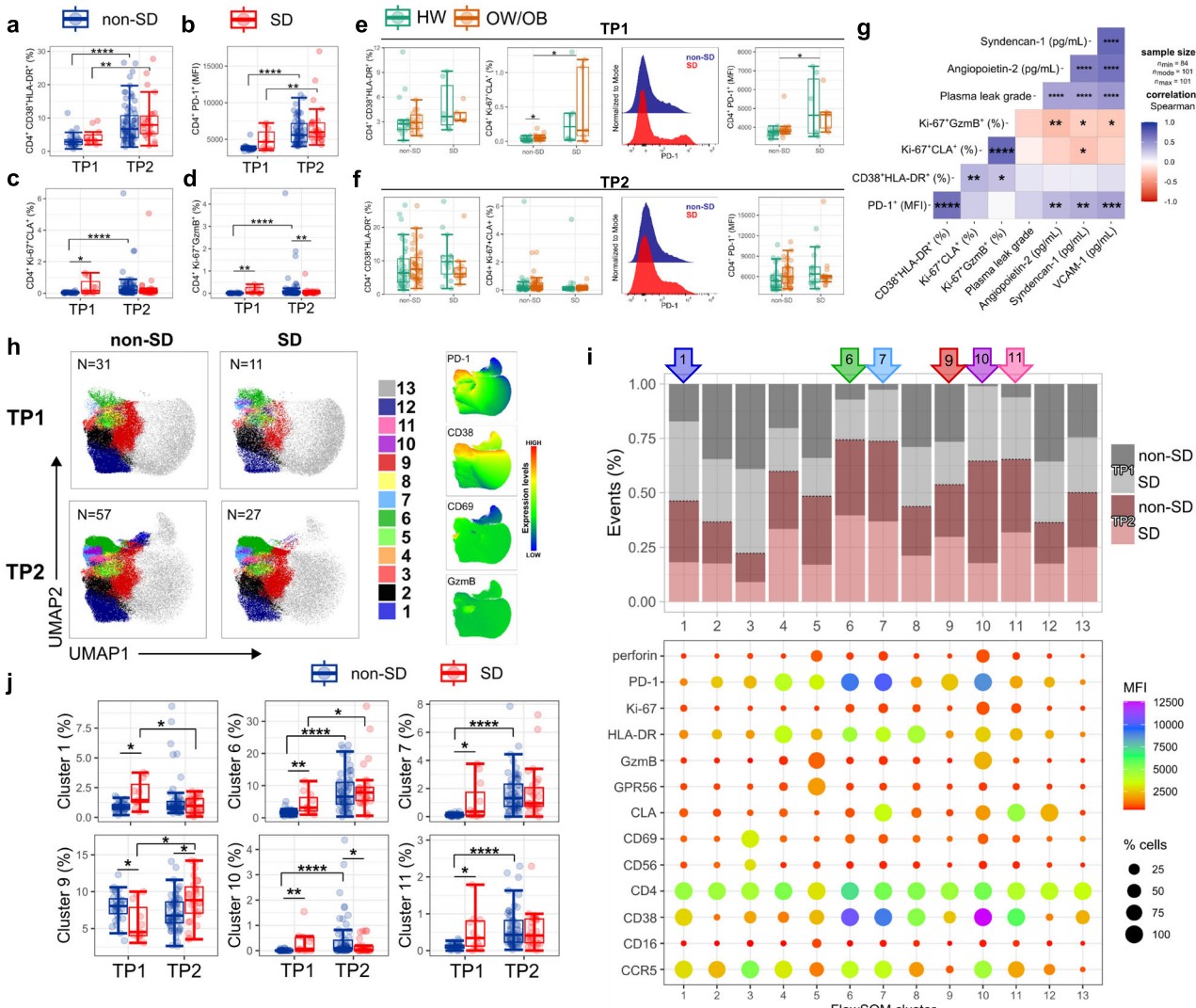

**Fig. 2 | Increased PD-1⁺ CD4⁺ T-cells in SD. a** Percentage of CD4⁺ T-cells co-expressing CD38 and HLA-DR. **b** PD-1 expression in CD4⁺ T-cells expressed as Median Fluorescence Intensity (MFI). **c, d** Percentage of CD4⁺ T-cells co-expressing Ki-67 and CLA (**c**) and Ki-67 and GzmB (**d**). **e, f** Percentages of CD38⁺ HLA-DR⁺ and Ki67⁺ CLA⁺ CD4⁺ T-cells and PD-1 MFI levels in CD4⁺ T-cells at TP1 (**e**) and TP2 (**f**). **a–f** TP1, $N = 42$; TP2: $N = 104$. PD-1 staining is shown for two representative patients with non-severe dengue (non-SD) or severe dengue (SD). Data for patients with non-severe dengue (non-SD), severe dengue (SD), healthy weight (HW) and overweight/obesity (OW/OB) are shown in blue, red, green and orange circles, respectively. **g** Correlation of CD4⁺ T-cell subsets with clinical parameters/biomarkers at TP2 ($N = 104$), using Spearman's rank correlation test with FDR correction. **h** UMAP

plots with FlowSOM clusters (1, 6, 7, 9–11) visualised in non-SD and SD patient groups at TP1 ($N = 42$) and TP2 ($N = 84$) and expression levels of PD-1, CD38, CD69, and GzmB. **i** Stacked bar chart showing the frequency (*y*-axis) of each cluster in the patient groups with the bubble graph representing MFI levels (colour scale) and cell frequencies (size). The clusters are indicated on the *x*-axis; arrows highlight selected clusters on the top of the graph. **j** Frequency of the FlowSOM clusters showed in (**h**, **i**) (1, 6, 7, 9–11) at TP1 ($N = 42$) and TP2 ($N = 84$) within non-SD and SD patient groups. The middle line in each box represents the median with IQR. Error bars represent max/min value ± 1.5*IQR. *$p < 0.05$; **$p < 0.01$; ***$p < 0.001$; ****$p < 0.0001$ calculated by one-way ANOVA with Benjamini–Hochberg correction. Source data are provided as a Source Data file.

CD4⁺ T-cell activation and PD-1 expression do not differ between patients with HW or OW/OB, while non-SD OW/OB patients display a modest increase in Ki-67⁺CLA⁺ DENV-specific CD4⁺ T-cells at TP1 compared to HW non-SD (Fig. 2e, f). Plasma levels of the markers of endothelial dysfunction angiopoietin-2, syndecan-1, and Vascular cell adhesion molecule 1 (VCAM-1) show a moderate positive correlation with PD-1 expression in CD4⁺ T-cells (Fig. 2g, shown for TP2). Conversely, these endothelial dysfunction-related markers correlated negatively with the frequency of Ki-67⁺GzmB⁺ and Ki-67⁺CLA⁺ CD4⁺ T-cells. These data show a correlation of PD1 + CD4⁺ T-cells with SD and of a cytotoxic CD4⁺ T-cell response with uncomplicated dengue.

To better understand the combinatorial expression of phenotypic/functional markers in CD4⁺ T-cells, we performed unsupervised dimensionality reduction analysis using uniform manifold

approximation and projection (UMAP) and the FlowSOM clustering (Fig. 2h–j). FlowSOM analyses of concatenated flow cytometry standard (FCS) files at TP1/TP2 from a total of 126 patient samples identify 13 phenotypically distinct CD4⁺ T-cell clusters based on the expression of the analysed markers. The six clusters that were most distinctive in non-SD and SD patients at TP1 and TP2 are characterised by high expression of PD-1, CD38, CD69 and GzmB (Fig. 2h–j, clusters 1, 6, 7 and 9–11). The frequencies of the 13 clusters within non-SD and SD patients at TP1/TP2, and expression of markers in each cluster are shown in Fig. 2i. Clusters 1, 6, 7 and 10–11 are highly expressed in SD versus non-SD patients at TP1 and include activated CD4⁺ T-cells with low cytotoxic potential (cluster 1, 6, 7) as well as high PD-1 and CCR5 expression (clusters 6, 7). Moreover, CD4⁺ T-cells in cluster 7 display increased expression of CLA and Ki-67,

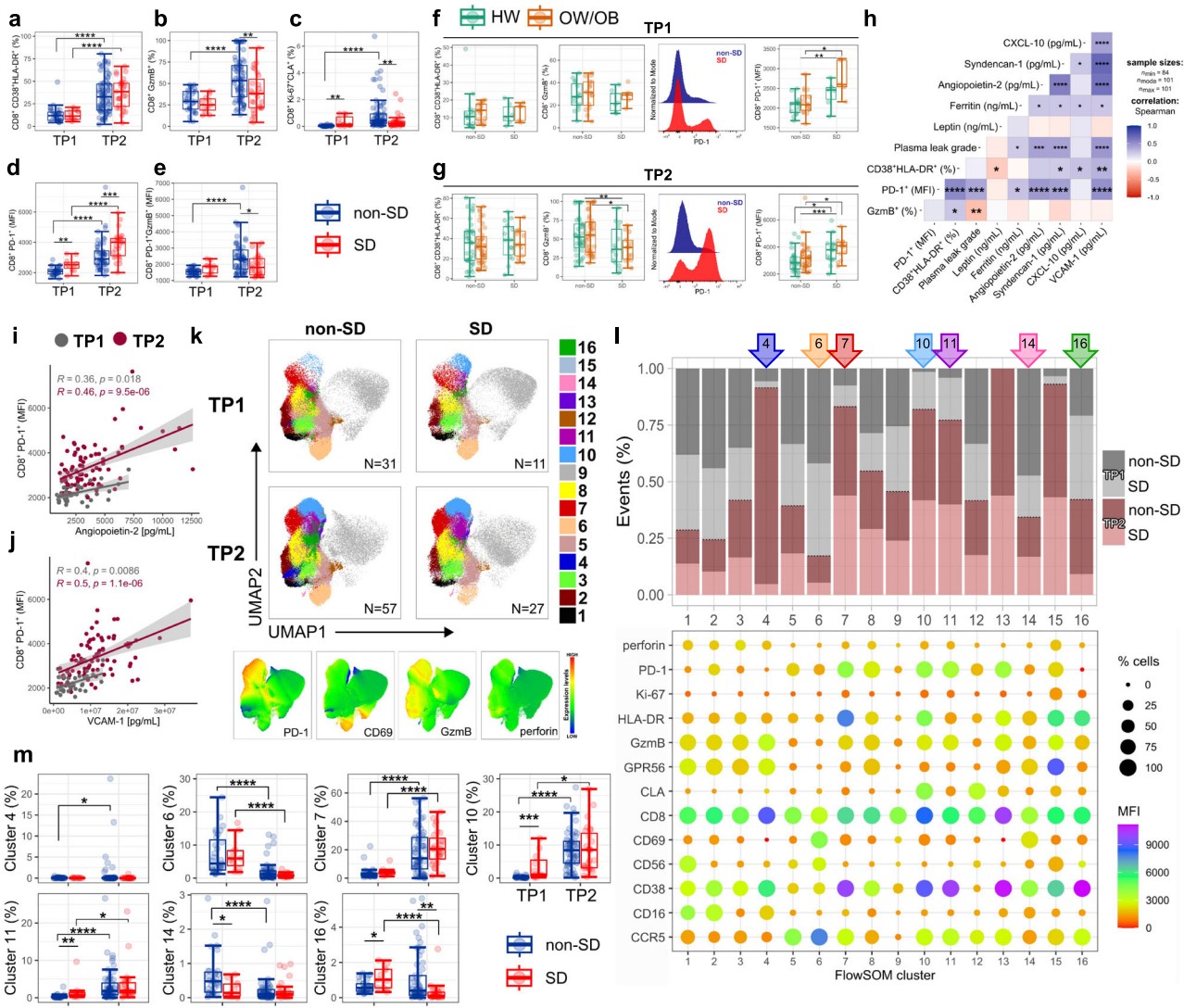

**Fig. 3 | Altered PD-1 and GzmB expression in CD8⁺ T-cells of SD patients.**
**a**–**e** Percentage of CD8⁺ T-cells expressing CD38 and HLA-DR (**a**), GzmB (**b**) and Ki-67 and CLA (**c**). PD-1 expression levels in CD8⁺ T-cells expressed as Median Fluorescence Intensity (MFI) (**d**). **e** Expression levels of GzmB (MFI) in PD-1⁺ CD8⁺ T-cells. Data is shown at TP1 ($N = 42$) and TP2 ($N = 104$). **f**, **g** Percentages of CD38⁺ HLA-DR⁺ and GzmB⁺ CD8⁺ T-cells and PD-1 MFI levels in CD8⁺ T-cells at TP1 (**f**; $N = 42$) and TP2 (**g**; $N = 104$). PD-1 staining is shown for two representative patients with non-SD or SD. Data for patients with non-severe dengue (non-SD), severe dengue (SD), healthy weight (HW) and overweight/obesity (OW/OB) are shown in blue, red, green and orange circles, respectively. **h** Correlation of CD8⁺ T-cell subsets with clinical parameters/biomarkers at TP2 ($N = 104$), using Spearman's rank correlation test with Benjamini–Hochberg correction. **i**, **j** Single correlations of CD8⁺ T-cells

expressing PD-1 with plasma levels of angiopoietin-2 (**i**) and VCAM-1 (**j**) at TP1 ($N = 42$) and TP2 ($N = 104$). **k** UMAP plots with FlowSOM clusters (4, 6, 7, 10, 11, 14, 16) visualised in non-SD and SD patient groups at TP1 ($N = 42$) and TP2 ($N = 84$) and expression levels of PD-1, CD69, GzmB, and perforin. **l** Stacked bar chart showing the frequency (y-axis) of each cluster in the patient groups, with the bubble graph representing MFI levels (colour scale) and cell frequencies (size). The clusters are indicated on the x-axis; arrows highlight selected clusters on the top of the graph. **m** Frequency of the FlowSOM clusters shown in (**k**, **l**) (4, 6, 7, 10, 11, 14, 16) at TP1 ($N = 42$) and TP2 ($N = 84$) within non-SD and SD patient groups. The middle line in each box represents the median with IQR. Error bars represent max/min value ± 1.5*IQR. *$p < 0.05$; **$p < 0.01$; ***$p < 0.001$; ****$p < 0.00001$ by one-way ANOVA with Benjamini–Hochberg correction. Source data are provided as a Source Data file.

suggesting this cluster of cells could represent CD4⁺ T-cells responding to DENV.

In summary, manual gating and unsupervised analyses show increased CD4⁺ T-cell expression of PD-1 in SD compared to non-SD patients, and expansion of PD-1⁺ CD4⁺ T-cell populations with low cytotoxic potential, with these CD4⁺ T-cell features correlating with endothelial markers of SD.

### Altered PD-1 and GzmB expression in CD8⁺ T-cells in SD
Similarly to CD4⁺ T-cells, CD8⁺ T-cell activation increases from TP1 to TP2 but does not differ in patients across disease severities (Fig. 3a). In non-SD, the frequencies of cytotoxic GzmB⁺ CD8⁺ T-cells increase from

TP1 to TP2, mirroring CD8⁺ T-cell activation. However, in SD patients the frequencies of GzmB⁺ CD8⁺ T-cells appear to uncouple from CD8⁺ T-cell activation and are decreased compared to non-SD patients at TP2 (Fig. 3b). Ki-67⁺CLA⁺ CD8⁺ T-cells display similar expansion kinetics to their CD4⁺ T-cell counterparts, with SD patients displaying a higher frequency of these cells at TP1 compared to non-SD patients, and an opposite trend is observed at the later time point (Fig. 3c). PD-1 levels also increase in time from TP1 to TP2 with SD patients displaying significantly higher expression levels compared to non-SD patients at both time points (Fig. 3d). Similar to what we observed in total CD8⁺ T-cells, PD-1⁺ CD8⁺ T-cells of SD patients display decreased GzmB expression compared to non-SD patients (Fig. 3e). Data stratified by

patient BMI status and disease severity shows similar frequencies of activated and GzmB[+] CD8[+] T-cells in HW and OW/OB patients, but a trend towards increasingly higher PD-1 expression levels from non-SD HW, non-SD OW/OB, SD HW and SD OW/OB patients at both time points, suggesting that in dengue PD-1 expression in CD8[+] T-cells may be exacerbated by high BMI (Fig. 3f, g). PD-1 expression in CD8[+] T-cells shows a strong positive correlation with plasma leakage grade, markers of endothelial dysfunction and with the dengue severity-related markers (ferritin and CXCL-10/IP-10) at both time points (Fig. 3h shown for TP2; Fig. 3i, j: angiopoietin-2 and VCAM-1, shown for TP1 and TP2). Conversely, GzmB expression in CD8[+] T-cells negatively correlates with plasma leakage and endothelial dysfunction, suggesting a protective role of cytotoxic CD8[+] T-cells in dengue.

Unsupervised analyses using UMAP and FlowSOM clustering of concatenated FCS files at TP1/TP2 from 126 patient samples reveal 16 distinct CD8[+] T-cell clusters based on the combinatorial expression of the analysed markers (Fig. 3k–m). The clusters with the most distinct representation between non-SD and SD patient groups at TP1 and TP2 fall within areas of highest expression of PD-1, CD69, GzmB and perforin (Fig. 3k–m; clusters 4, 6, 7, 10, 11, 14 and 16). Clusters 10, 11 and 16 are present at higher frequencies in SD compared to non-SD patients at TP1 and contain CD8[+] T-cells with high PD-1 and CCR5 expression. Cells from cluster 10 are CLA[+]Ki-67[+] and may represent CD8[+] T-cells responding to DENV. Cluster 4 is detected only in non-SD patients, although in a minor proportion of these patients ($N = 7/42$), and contains moderately activated cells, with high cytotoxic potential (GzmB[+], GPR56[+]) and low PD-1 and CCR5 expression. Clusters 6 and 7 contain cells that are respectively increased at TP1 and TP2, suggesting they represent T-cell populations at early and late differentiation stages. Accordingly, CD8[+] T-cells in cluster 6 cells express high levels of the early activation marker CD69 and CCR5, while those in cluster 7 express the activation markers HLA-DR/CD38, PD-1 and display increased cytotoxic potential. Collectively, these data demonstrate decreased PD-1 and GzmB expression in CD8[+] T-cells from SD patients, which strongly correlates with clinical markers of dengue disease severity.

**Altered DENV NS3-specific T-cell response in SD**

To address whether disease severity and BMI associate with altered DENV-specific T-cell responses, we evaluated CD4[+] and CD8[+] T-cell responses after a brief stimulation of PBMCs with or without overlapping peptides spanning the immunodominant NS3 protein[29] or with PMA/ionomycin as a positive control. Intracellular cytokine staining (ICS) was performed to measure production of IFN-γ, TNF-α, IL-2, MIP-1β, and CD107a, an indirect marker of degranulation by flow cytometry (Fig. 4a). To investigate the phenotypic features of DENV-specific T-cells, we co-stained cells with antibodies targeting markers of T-cell differentiation (CD95) and activation/exhaustion (PD-1). For these analyses, we selected patients with secondary DENV-2 infection. The frequencies of DENV2 NS3-specific CD4[+] and CD8[+] T-cells, defined as cytokine[+] and/or CD107a[+] T-cells, are higher in SD compared to non-SD patients at TP1, with an opposite trend at the later time point, similarly to the Ki-67[+] CLA[+] DENV-specific T-cells (Fig. 4b; gating strategy in Supplementary Fig. S3a). To investigate whether there was a skewed T-cell response to DENV serotypes potentially encountered during the primary infection (OAS), driven by expansion of pre-existing memory T-cells, we tested recognition of NS3 peptide pools from all four DENV serotypes. Cytokine production by CD4[+] and CD8[+] T-cells upon recognition of NS3 peptide pools from DENV1–4 serotypes is similar, but responses are overall lower in SD compared to non-SD patients at TP2 (Fig. 4c and Supplementary Fig. S3b, c). These data suggest broad cross-reactive T-cell recognition of NS3 peptides from all 4 DENV serotypes with no preferential skewing for a specific serotype, and increased magnitude of NS3-specific T-cell responses in non-SD patients at the time of viral clearance.

We next investigated whether T-cell cytokine profiles differ across patient groups (Fig. 4d, e). DENV2 NS3-specific T-cells are mainly monofunctional (i.e., expressing a single cytokine/function) with a minor proportion of polyfunctional cells expressing 2–5 functions (Supplementary Fig. S4). We therefore focused our analysis on T-cells producing single functions. Early in infection, at TP1, DENV2 NS3-specific CD4[+] T-cells mainly produce IFN-γ and CD107a, with a stepwise increase of IFN-γ production observed from non-SD to SD HW and SD OW/OB patients (Fig. 4d, top panel). DENV2 NS3-specific CD8[+] T-cells display a similar stepwise increase of IFN-γ production from non-SD to SD HW and SD OW/OB patients at TP1 (Fig. 4e, top panel). At this time point, SD patients also display increased frequencies of degranulating CD107a[+] DENV2 NS3-specific CD8[+] T-cells. At TP2, IFN-γ/CD107a production by DENV2 NS3-specific CD4[+] and CD8[+] T-cells was comparable across groups (Fig. 4d, e, bottom panels). Production of IL-2, TNF-α and MIP-1β by NS3-specific CD4[+] and CD8[+] T-cells was also largely similar across groups at both time points. These findings are in line with the higher frequencies of Ki67[+]CLA[+] T-cells shown above and total cytokine[+]/CD107a[+] T-cells (Fig. 4b) at TP1 in SD compared to non-SD patients.

In line with previous studies[16], at TP1, DENV2 NS3-specific T-cells in SD patients contained higher percentages of cytokine-producing/CD107a-negative and lower percentages of cytokine-negative/CD107a-positive cells, suggesting a skewing of T-cell responses towards cytokine production in SD and conversely towards cytotoxicity in non-SD patients (Fig. 4f). As SD patients display higher viraemia early in infection[30], we asked whether T-cells may be undergoing excessive antigen-driven activation leading to antigen-induced cell death. A higher proportion of NS3 DENV2-specific CD4[+] and CD8[+] T-cells express PD-1 at TP1 in SD patients, while these cells are largely absent in non-SD patients (Fig. 4g). PD-1[+] NS3 DENV2-specific CD4[+] and CD8[+] T-cells from SD patients express high levels of the death receptor CD95 compared to their PD-1[−] counterparts, suggesting these cells may be undergoing cell death, possibly due to excessive antigenic stimulation. At TP2, CD95 levels of PD-1[+] cells are significantly decreased from TP1, but they remain higher compared to PD-1[−] cells (Fig. 4h). Collectively, these data suggest that SD patients display higher frequencies of NS3 DENV-specific T-cells early in infection, which are skewed towards cytokine production, express PD-1 and CD95 and may be prone to apoptosis.

**Elevated T-cell co-inhibitory receptors in SD**

We next asked whether T-cells in SD patients express other co-inhibitory receptors beyond PD-1. To this end, we analysed expression of five co-inhibitory receptors associated with T-cell exhaustion (CTLA-4, LAG-3, TIM-3, PD-1, and TIGIT)[31]. At TP1, SD patients display significantly increased frequencies of TIGIT[+] and TIM-3[+] CD4[+] T-cells and LAG-3[+] CD8[+] T-cells and a trend towards increased frequencies of CD4[+]/CD8[+] T-cells expressing all co-inhibitory receptors analysed (Fig. 5a, b). Similar to what we observed for PD-1, the frequencies of CD4[+]/CD8[+] T-cells expressing these inhibitory receptors correlate positively with endothelial dysfunction and severity-related plasma markers (syndecan-1, VCAM-1, and ferritin) (Fig. 5c, d). In SD patients, there was a larger frequency of CD4[+] and CD8[+] T-cells co-expressing multiple co-inhibitory receptors compared to non-SD patients, with TIGIT and PD-1 being the most highly expressed (Fig. 5e), suggesting these T-cells may be exhausted. To gain further insights into the features of T-cells expressing multiple co-inhibitory receptors we established a flow cytometry panel which included six co-inhibitory receptors (CTLA-4, LAG-3, TIM-3, PD-1, TIGIT and LILRB1), activation/proliferation markers (ICOS, CD25, Ki-67) and markers for T-cells (CD3, CD4, CD8), NK-cells (CD16, CD56) and Tregs (CD25, FOXP3). UMAP and FlowSOM clustering analyses identified 15 distinct CD4[+] and CD8[+] T-cell clusters; clusters showing significant differences in SD versus non-SD patients are colour-coded in the UMAP (Fig. 5f: CD4[+] T clusters 13–15; Fig. 5g: CD8[+]

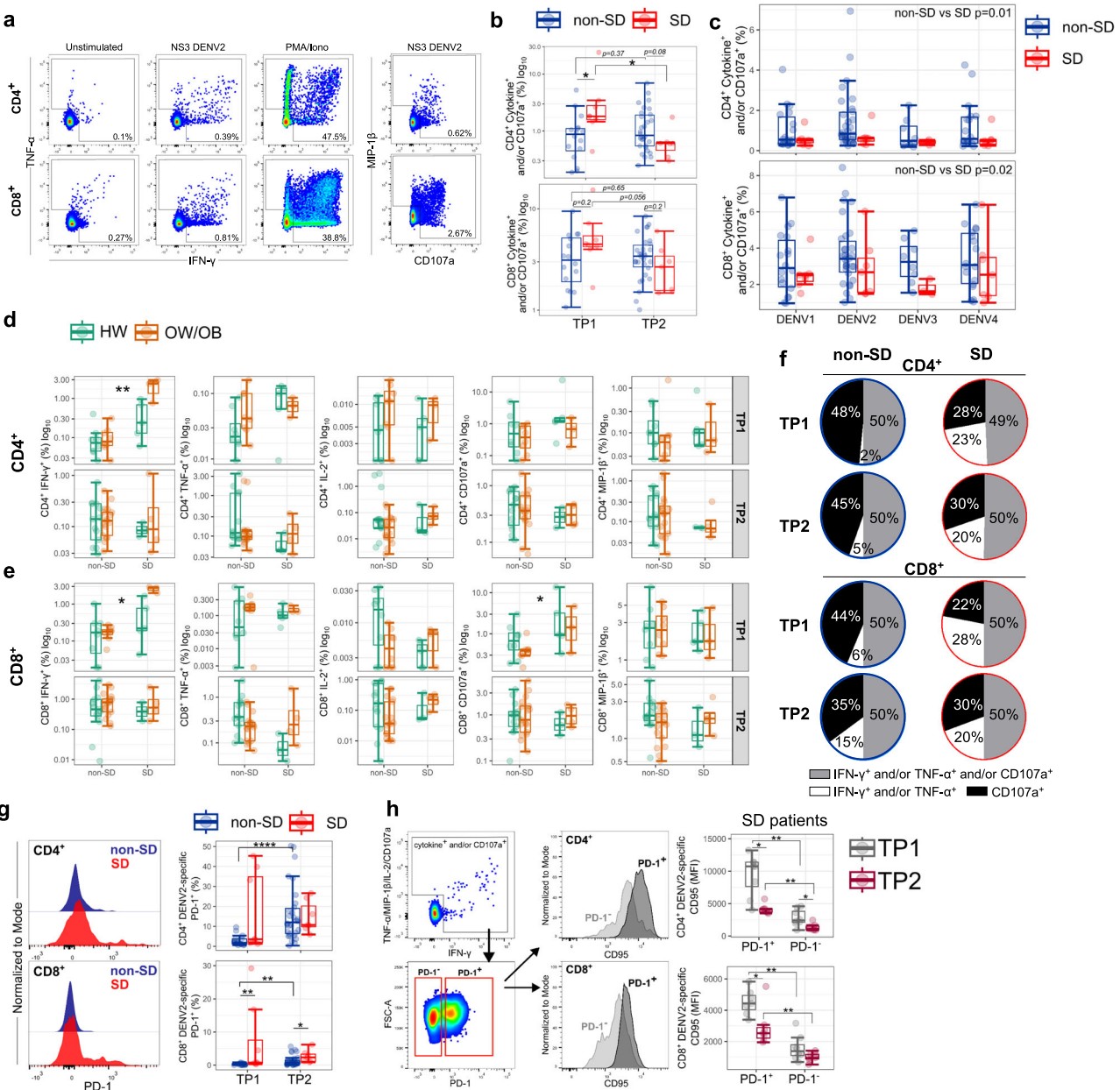

**Fig. 4 | Altered DENV NS3-specific T-cells in SD. a** Representative flow cytometry plots showing IFN-γ, TNF-α, MIP-1β, and CD107a production by CD4⁺ and CD8⁺ T-cells from a non-SD patient (day 6 of illness onset) after stimulation with NS3 DENV2 peptides, PMA/ionomycin, or unstimulated (DMSO). All data shown here were obtained in PBMCs of patients with DENV2 infection. **b** CD4⁺ and CD8⁺ T-cell responses shown as log10 of cytokine⁺ (IFN-γ and/or TNF-α and/or IL-2 and/or MIP-1β) and/or CD107a ⁺ cells at TP1 (*N* = 24) and TP2 (*N* = 37) after stimulation with NS3 DENV2 peptides. **c** CD4⁺ and CD8⁺ T-cell responses shown as cytokine⁺ and/or CD107a⁺ at TP2 (*N* = 37) after stimulation with NS3 DENV1–4 peptide pools. Data for patients with non-severe dengue (non-SD) and severe dengue (SD) are shown in blue and red circles, respectively. **d**, **e** Production of each cytokine and CD107a is shown for CD4⁺ (**d**) and CD8⁺ (**e**) T-cells in HW or OW/OB patients with non-SD and SD at TP1 (*N* = 24) and TP2 (*N* = 37) following NS3 DENV2 peptide stimulation. Data for patients with healthy weight (HW) and overweight/obesity

(OW/OB) are shown in green and orange circles, respectively. **f** Cytokine⁺ and/or CD107a⁺ T-cells were divided into three groups based on their properties: degranulation only (CD107a), degranulation and cytokine production (IFN-γ and/or TNF-α and/or CD107a), and cytokine production only (IFN-γ and/or TNF-α). Pie charts show the average percentages of the cytokine⁺ and/or CD107a⁺ cells in total responding T-cells from non-SD and SD patients. **g** Representative histogram and boxplots showing PD-1 expression in DENV2-specific CD4⁺ and CD8⁺ T-cells at TP1 (*N* = 24) and TP2 (*N* = 37). **h** Gating strategy and representative flow cytometry plots of CD95 and PD-1 expressing T-cells. CD95 MFI levels in PD-1⁺ and PD-1⁻ CD4⁺ and CD8⁺ T-cells are shown for SD patients (TP1, *N* = 8; TP2, *N* = 7). The middle line in each box represents the median with IQR. Error bars represent max/min value ± 1.5*IQR. *p < 0.05; **p < 0.01; ***p < 0.001; ****p < 0.0001 by one-way ANOVA with Benjamini–Hochberg correction. Source data are provided as a Source Data file.

T clusters 12–14). CD4⁺ T-cells in clusters 13–15 are largely detectable only in SD patients. Cluster 13 contains proliferating ICOS⁺ CD4⁺ T-cells co-expressing all six co-inhibitory receptors as well as CD16 and CD56. Cells in cluster 14 co-express four inhibitory receptors, although at lower levels compared to cluster 13, and they lack expression of Ki-67. Cluster 15 contains proliferating CD4⁺ CD25⁺FOXP3⁺ regulatory T-cells

(Tregs) expressing TIGIT and CD56 (Fig. 5f). Similarly, SD patients display increased levels of non-proliferating CD8⁺ T-cells expressing co-inhibitory receptors TIGIT, TIM-3, and PD-1 or ICOS, LILRB1, CTLA-4, LAG-3, PD-1 and TIGIT (Fig. 5g, respectively clusters 13 and 14). Interestingly, CD8⁺ T-cells in the latter cluster also express FOXP3, suggesting they may represent CD8⁺ Tregs.

Collectively, these data show higher frequencies in SD compared to non-SD of responding (activated and/or proliferating) CD4[+] and CD8[+] T-cells co-expressing inhibitory receptors, with the highest PD-1 and TIGIT expression, as well as increased Treg populations in SD compared to non-SD.

T-cell function is closely linked with cellular metabolism, and the latter is shown to be altered in obesity. We therefore asked whether T-cells expressing co-inhibitory receptors have altered metabolic profiles in SD and in OW/OB, using PD-1 as a representative marker. For these analyses, we selected a total of 17 patients; all except one patient had a secondary DENV2 infection (non-SD: $N = 11$; SD: $N = 6$, including $N = 8$ HW and $N = 9$ OW/OB; illness days 5–8). The metabolic profiles of PD-1[+]/PD-1[−] CD4[+] and CD8[+] T-cells were assessed by measuring expression by flow cytometry of four key metabolic enzymes/components involved in ATP biosynthesis (ATP5A), fatty acid oxidation, FAO (CPT1A), and glycolysis (HK1 and GLUT1). PD-1[+] CD4[+] and CD8[+] T-cells from OW/OB patients display increased expression levels of GLUT1 and CPT1A compared to their HW counterparts (Fig. 5h), with levels being similar in non-SD and SD patients. PD-1[+] CD4[+] and CD8[+] T-cells display elevated metabolic activity compared to their PD-1[−] counterparts, based on expression of HK1, GLUT1, ATP5A and CPT1A, with statistically significant differences only within non-SD patients (Fig. 5i). Similar results were obtained using the SCENITH (Single Cell Energetic metabolism by profiling translation inhibition) method[32,33] (Supplementary Fig. S5). These data suggest that PD-1[+] CD4[+] and CD8[+] T-cells are engaging in glycolysis, FAO and ATP biosynthesis to support their metabolic demands, more so in OW/OB compared to HW patients.

We next asked whether PD-1 expression is playing a role in inhibiting antiviral T-cell effector functions in dengue, specifically their cytotoxic potential, which appears to be impaired. To address this, we performed an overnight stimulation with NS3 DENV peptides (matched to the serotype of infection) in the presence of anti-PD-1 and/or anti-PD-L1 blocking antibodies or an isotype control and measured cytokine production and GzmB/perforin production of CD4[+] and CD8[+] T-cells. Prior to anti-PD(L)1 blockade, we evaluated the ex vivo expression of PD-1 and PD-L1 in CD8[+] T-cells (Fig. 5j). After stimulation and blockade, we measured expression of cytotoxic mediators and cytokine production. While the effects on DENV-specific T-cells were difficult to evaluate due to the limiting number of these cells, overall PD-1 and anti-PD-1/PDL-1 blockade led to increased frequencies of GzmB[+] perforin[+] CD8[+] T-cells in some patients, with variability across patients (Fig. 5k). While PD-1 and PD-1/PDL-1 blockade did not affect the frequencies of CD8[+] T-cells producing each analysed cytokine, there was a trend towards increased T-cell polyfunctionality defined as co-expression of CD107a, IFN-γ and TNF-α (Fig. 5l).

## Impaired NK-cell responses in SD

NK-cell function is governed by the balance of signals from cell surface activating and inhibitory receptors[34]. We therefore assessed NK-cell expression of activating (NKG2D, NKp46) and inhibitory (LILRB1, NKG2A, PD-1, PDL-1, TIGIT and LAG-3) receptors as well as markers of activation (CD69), proliferation (Ki-67), differentiation (CD57, NKG2C) and cytotoxic potential (GzmB, perforin) in samples from 23 patients (non-SD: $N = 11$; SD: $N = 12$) at TP1. To determine the features of NK-cells that are responding to DENV infection, we analysed proliferating Ki-67[+] NK-cells, herein defined as responding NK-cells (Fig. 6a). Data is shown for total responding NK-cells (findings were similar for NK CD56[dim] and NK CD56[bright] cells, data not shown). In line with our previous findings, responding NK-cells in dengue infection are predominantly immature CD57[−]NKG2C[−] cells[35], with this being more pronounced in SD patients (Fig. 6b). Interestingly, SD patients display lower frequencies of CD57[+]NKG2C[+] "memory" NK-cells. Responding NK-cells from SD patients are less activated (CD69) and displayed decreased cytotoxic potential (GzmB, perforin) and decreased expression of activating

receptors NKG2D and NKp46, compared to those from non-SD patients (Fig. 6c). Conversely, the expression levels of NK-cell co-inhibitory receptors LILRB1, NKG2A, PD-L1, TIGIT and PD-1 and the frequencies of LAG-3[+] NK-cells are higher in SD compared to non-SD patients (Fig. 6d). Collectively, these data suggest a potential impairment in the NK-cell response in SD patients.

Plasma leakage grade, endothelial dysfunction (angiopoietin-2, sydnecan-1, VCAM-1) and the severity-related biomarker (ferritin) positively correlate with expression of NK-cell inhibitory markers (LILRB1, NKG2A, PD-L1) and negatively correlate with the frequencies of activated and cytotoxic NK-cells (CD69[+], NKG2D[+], GzmB[+] perforin[+]; Fig. 6e). Plasma levels of soluble MICB, the ligand of NKG2D, correlate inversely with ferritin, suggesting that MICB-NKG2D interactions associate with less severe disease. Accordingly, NKG2D[+] NK-cell and CD8[+] T-cells are significantly increased in non-SD compared to SD patients (Fig. 6d and Supplementary Fig. S6). These analyses highlight the association of different dengue severity-related markers with NK-cell impairment.

Unsupervised UMAP/FlowSOM analyses show similar findings. SD patients display increased clusters of NK-cells expressing the inhibitory receptors LILRB1 and NKG2A and non-proliferating NK-cells expressing NKp44, NKp46 and NKG2D and NKG2A (Fig. 6f–h, respectively clusters 8 and 10). Conversely, activated NK-cells with a more mature phenotype expressing NKG2C and CD57 are present at higher frequencies in non-SD compared to SD patients (Fig. 6f–h, cluster 1). In a second flow cytometry panel, the key difference between non-SD and SD patients is observed in cluster 8, with a significantly increased percentage in SD patients (Fig. 6i–k). These NK-cells are highly activated and proliferating and express high levels of four inhibitory receptors (LAG-3, TIGIT, PD-1, and PD-L1). Clusters 1 and 3 represent activated NK-cells with increased cytotoxic capacity, which show a trend of decreased frequency in SD compared to non-SD patients. In the LDA analysis, the phenotypic and functional features of NK-cells early in infection (TP1) can discriminate patients with SD from those with non-SD (Fig. 6l). NK-cell features are largely overlapping in HW, OW and OB patients (Fig. 6m), although manual gating analyses revealed decreased GzmB[+] perforin[+] cells and LILRB1 expression in OW/OB compared to HW non-SD patients (Supplementary Fig. S7).

We next determined whether NK-cells from SD patients were altered in their effector function compared to non-SD patients. To this end, NK-cells were assessed for natural cytotoxicity (degranulation assessed as CD107a expression) and production of IFN-γ, TNF-α and MIP-1β after co-culture with K562 target cells with or without cytokine stimulation (IL-12/18). In response to stimulation with K562 cells, NK-cells from SD and non-SD patients produce similar amounts of IFN-γ and no TNF-α or MIP-1β (Fig. 6n–q). However, in this condition, NK-cell degranulation (CD107a) is significantly decreased in NK-cells from SD patients. IL-12/18 stimulation induces NK-cell production of IFN-γ and boosts all four measured functions when combined with K562 stimulation, but it was not able to restore NK-cell degranulation in SD to the levels observed in non-SD patients.

Collectively, our data demonstrate phenotypically and functionally altered NK-cells early in infection prior to or at early onset of SD.

## Impaired type-I IFN signalling in SD

We next investigated whether defects in the innate IFN response could underlie T-cell and NK-cell impairment in SD. To achieve this, we performed targeted gene expression analysis by single-cell RNA-sequencing (scRNA-seq, BD Rhapsody Immune Response Panel HS + a custom panel with an additional 145 genes, Supplementary Table S2) of PBMCs from 24 dengue patients at TP1 ($N = 12$ SD; $N = 12$ non-SD, with each group including 6 HW and 6 OW/OB patients matched by sex and age). scRNA-seq captured gene expression of circulating CD4[+]/CD8[+] T-cells, NK-cells, MAIT cells, γδ T-cells, B-cells, plasmablasts, monocytes, and dendritic cells (Fig. 7a). Plasmablast frequency was increased in SD

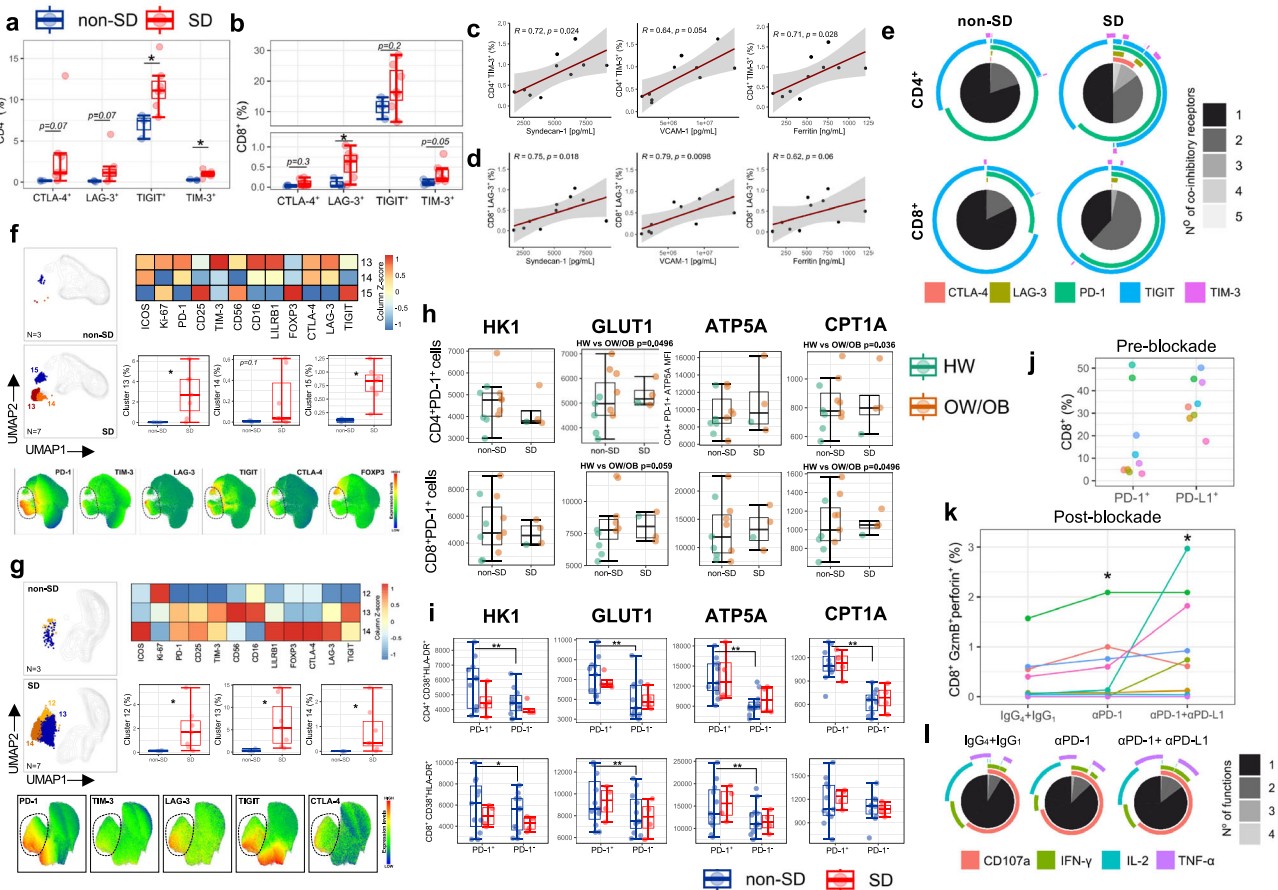

**Fig. 5 | Elevated T-cell co-inhibitory receptors in SD. a**, **b** Percentage of co-inhibitory expression in CD4⁺ (**a**) and CD8⁺ (**b**) T-cells at TP1 in $N = 10$ patients with non-severe (non-SD; blue circles) or severe dengue (SD; red circles). **c**, **d** Correlation of the indicated markers with frequencies of TIM-3⁺ CD4⁺ T-cells (**c**) and LAG-3⁺ CD8⁺ T-cells (**d**). **e** Pie charts showing the number of co-inhibitory receptors simultaneously expressed by CD4⁺ and CD8⁺ T-cells in non-SD and SD patients at TP1 ($N = 11$). The different shades of grey represent the expression of 1 to 5 co-inhibitory receptors, and the outer arcs indicate the specific co-inhibitory receptors expressed as defined by Boolean gating. **f**, **g** UMAP plots showing selected Flow-SOM clusters of CD4⁺ (**f**) and CD8⁺ T-cells (**g**) at TP1 in non-SD and SD patients ($N = 11$). The expression level of each marker is shown in heatmap (Z-score) and UMAP plots; the frequency of each cluster is shown for non-SD and SD patients. **h** Expression levels of the indicated metabolic markers are shown as Median Fluorescence Intensity (MFI) in PD-1⁺ CD4⁺ and CD8⁺ T-cells of patients with healthy weight (HW; green circles) and overweight/obesity (OW/OB; orange circles)

experiencing non-SD and SD (day 5–8); $N = 15$. **i** Expression levels of the indicated metabolic markers shown as MFI in PD-1⁺ and PD-1⁻ activated (HLA-DR⁺ CD38⁺) CD4⁺ and CD8⁺ T-cells from non-SD and SD patients ($N = 17$). **j** Percentages of PD-1⁺ and PD-L1⁺ CD8⁺ T-cells in patient PBMCs prior to anti-PD-1/PDL-1 blockade ($N = 9$). **k** Frequency of GzmB⁺ perforin⁺ CD8⁺ T-cells after stimulation with NS3 DENV2 peptides in the presence of anti-PD-1/PDL-1 blocking antibodies or isotype controls. **l** Pie charts showing the number of functions simultaneously expressed by CD8⁺ T-cells following NS3 DENV2 peptide stimulation with or without blocking antibodies. The different shades of grey represent the expression of 1–4 functions, the outer arcs indicate the specific functions as defined by Boolean gating. The middle line in each box represents the median with IQR. Error bars represent max/min value ± 1.5*IQR. *$p < 0.05$; **$p < 0.01$; ***$p < 0.001$; ****$p < 0.0001$ calculated by one-way ANOVA with Benjamini–Hochberg correction. Source data are provided as a Source Data file.

compared to non-SD, consistent with our flow cytometry data (Supplementary Fig. S8a, b). Gene ontology (GO) over-representation analysis (ORA) of differentially expressed genes (DEG) revealed a blunted type-I IFN response in SD compared to non-SD patients in all cell types combined, as well as in CD4⁺ and CD8⁺ T-cells and NK-cells (Fig. 7b, adjusted $p < 0.05$, log2 FC < −0.25; Supplementary Fig. S9). Genes in the type-I IFN signalling pathway that were significantly downregulated in SD include IFN-induced transmembrane protein (*IFITM*) 1, 2 and 3, IFN regulatory factor family (*IRF*) 7 and 9, IFN-stimulated gene 15 (*ISG15*), IFN-induced helicase C domain-containing protein 1 (*IFIH1*), IFN inducible protein 16 (*IFI16*), 2′-5′-oligoadenylate synthetase 1 (*OAS1*), and signal transducer and activator of transcription (*STAT*) 1 and 2. IL-2R gamma subunit, which is common to different interleukin receptors, is also significantly downregulated in SD, suggesting decreased responsiveness to cytokines (Supplementary Table S3). To determine the extent of downregulation of type-I IFN signalling across different immune cells, we generated a type-I IFN

score based on expression of genes within the GO term "type-I interferon-mediated signalling pathway" (Fig. 7c). Type-I IFN signalling scores were decreased in PBMCs from SD versus non-SD patients and positively correlated with the plasma levels of IFN-α (Supplementary Fig. S10a–e). Decreased type-I IFN signalling was evident in CD4⁺ T naïve/central memory (TCM) and CD4⁺/CD8⁺ T effector memory (TEM) cells, naïve B-cells NK-cell, MAIT cells, proliferating lymphocytes and monocytes from patients with SD compared to non-SD (Fig. 7c). Cell-to-cell communication analyses revealed CXCL10-CXCR3 as the main molecular interaction that was downregulated across different cell types in SD compared to non-SD patients, in line with the IFN-induced expression of CXCL10 (Supplementary Fig. S11).

A limitation of our scRNA-seq data is the use of a probe-based approach, which limited our analyses to ~500 genes and precluded a comprehensive characterisation of the gene expression landscape. To expand our analyses and validate at the protein level the dampened type-I IFN signalling observed by scRNA-seq, we performed a

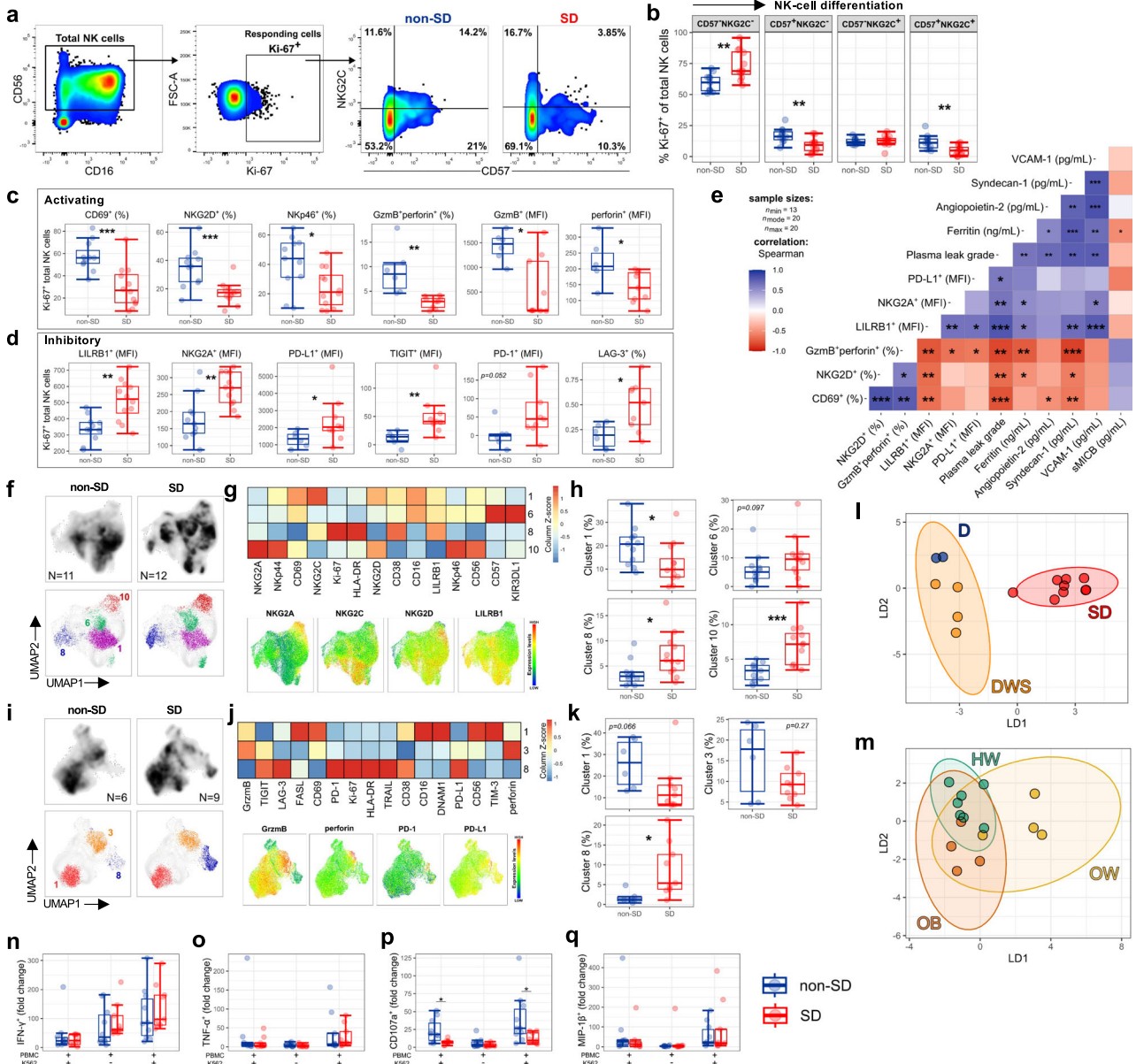

**Fig. 6 | Impaired NK-cell responses in SD. a** Gating strategy and representative flow cytometry plots of CD57 and NKG2C staining in non-severe (non-SD) and severe dengue (SD) patients at TP1. **b** Frequency of Ki-67⁺ NK-cells with differential expression of CD57 and NKG2C in non-SD (N = 11) and SD (N = 12) patients. **c, d** Expression of activating (**c**) and inhibitory (**d**) receptors in Ki-67⁺ NK-cells from non-SD and SD patients (**c** N = 23; **d** N = 15). **e** Correlation of NK-cell markers with clinical parameters/biomarkers using Spearman's rank correlation test with Benjamini–Hochberg correction. **f–h** UMAP and phenograph cluster analyses of NK-cells in non-SD and SD patients, with clusters 1, 6, 8 and 10 highlighted in colour (**f**). The expression levels of each marker are shown in the heatmap (z-score) and UMAP (**g**), and the frequency of each cluster is shown in non-SD and SD patients (**h**). **f–h** N = 23 (non-SD, N = 11; SD, N = 12). **i–k** UMAP and phenograph cluster analyses of NK-cells in non-SD and SD patients, with clusters 1, 3 and 8 highlighted in colour (**i**). The expression levels of each marker are shown in the

heatmap (z-score) and UMAP (**j**), and the frequency of each cluster is shown in non-SD and SD patients (**k**); **i, k** N = 15 (non-SD, N = 6; SD, N = 9). **l, m** Linear discriminant analysis at TP1 (N = 23); LD1 and LD2 were derived using all features shown in (**b–d, h, k**). Data points represent individual patients with dengue (D; blue circles), dengue with warning signs (DWS; orange circles), and SD (red circles) (**l**), and patients with healthy weight (HW; green circles), overweight (OW; yellow circles), and obesity (OB; orange circles). **m** Ellipses represent 95% confidence intervals. **n–q** NK-cell expression of cytokines (IFN-γ, TNF-α, MIP-1β) and CD107a after stimulation of PBMCs with K562, IL-12 + IL-18, or both (N = 19). Data are shown as fold change from unstimulated PBMCs. The middle line in each box represents the median with IQR. Error bars represent max/min value ± 1.5*IQR, *p < 0.05; **p < 0.01; ***p < 0.001; ****p < 0.0001 calculated by Wilcoxon test or one-way ANOVA with Benjamini–Hochberg correction. Source data are provided as a Source Data file.

quantitative Tandem Mass Tag (TMT) mass spectrometry-based proteomics analysis of PBMC samples from 11 dengue patients at TP1 (N = 6 non-SD; N = 5 SD). This holistic whole-proteome analysis identified >6700 proteins of which 1655 are differentially expressed between SD and non-SD patients (adjusted p < 0.05), with 280 displaying a >1.5-fold downregulation in SD (log2FC < −0.585); Fig. 7d; Supplementary data 1. Screening of the identified proteins against a list of 350 proteins

encoded by known interferon-stimulated genes (ISGs) identified 54 ISGs that are significantly dysregulated (adjusted p < 0.05, listed in Fig. 7e), 27 with a 1.5-fold decrease (log2FC < −0.585) and 12 with a 2-fold decrease (log2FC < −1, highlighted in Fig. 7d). GO ORA of differentially expressed proteins revealed a downregulation of the innate immune response in SD as well as other biological processes such as cell migration and adhesion (Fig. 7f).

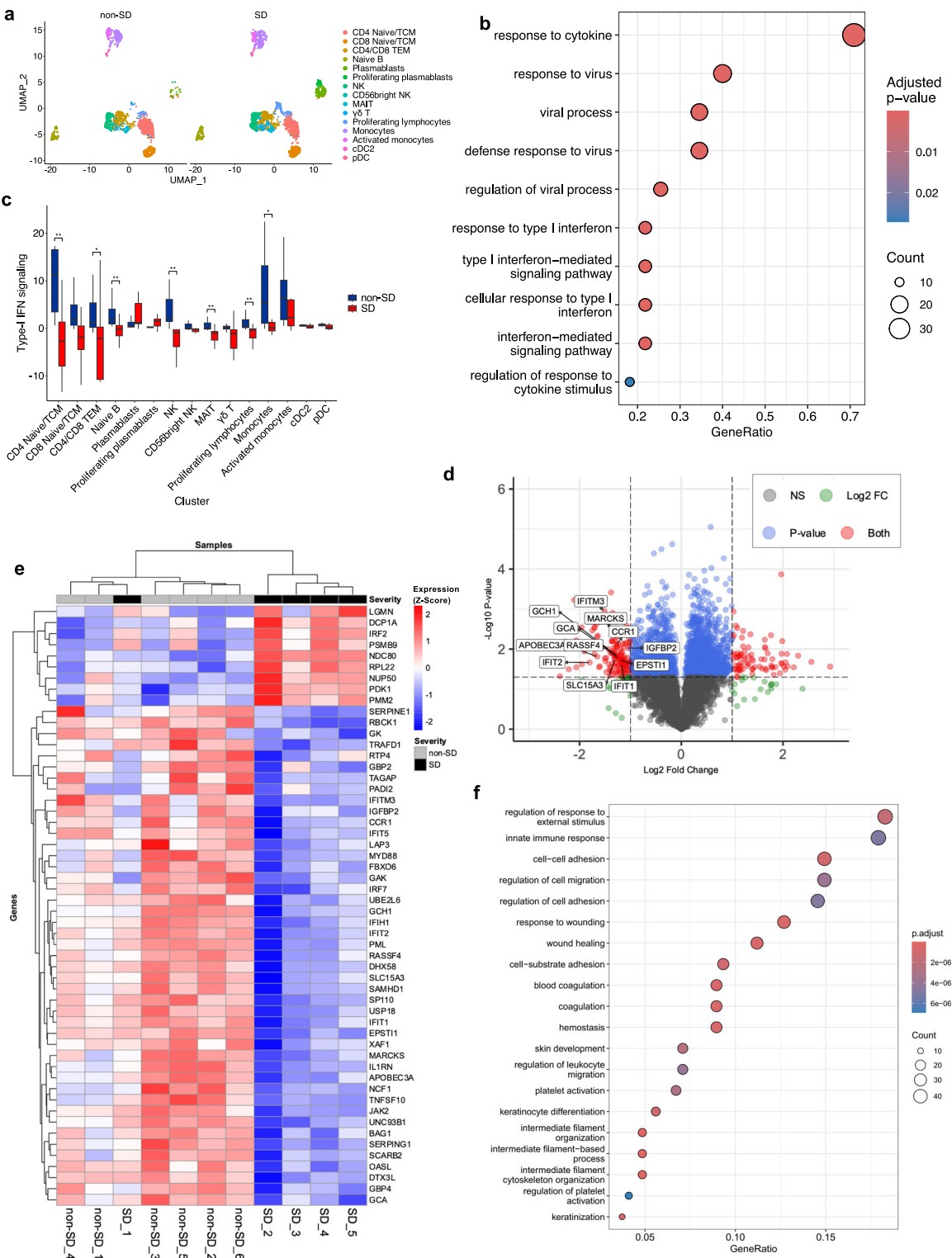

In summary, using transcriptional and proteomics approaches, we demonstrate decreased expression of genes and proteins involved in type-I IFN signalling in SD. Type-I IFNs mediate NK-cell activation in viral infection, including dengue[36] and are critical for the survival of activated CD8$^+$ T-cells[37]. Based on these data we propose dampened type-I IFN response as a potential mechanism underlying NK and T-cell dysfunction in SD.

## Discussion

The lack of validated correlates of protection and immunopathology for dengue represents a major challenge for the design of protective vaccines and host-directed therapeutics. Here we provide an in-depth analysis of T and NK-cell signatures associated with dengue disease severity and obesity. Our work provides unparalleled data encompassing phenotypic and functional profiles of T and NK-cells in a

**Fig. 7 | Impaired type-I IFN responses in SD.** ScRNA-seq (BD Rhapsody) analyses of PBMCs from 24 patients with non-severe dengue (non-SD; $N = 12$) and severe dengue (SD; $N = 12$) are shown in (**a**–**c**). **a** UMAP showing the manually gated immune cells coloured by cell types. **b** Gene Ontology (GO) Over-representation analysis of genes significantly downregulated (all cell types combined) in SD vs non-SD patients. *P*-values were adjusted for multiple comparisons using the Benjamini–Hochberg method. **c** Type-I IFN signalling module scores are shown for the different cell types. Single-cell scores were generated using genes from the "type I interferon-mediated signalling pathway" GO term (GO:0060337). Scores were summed for cells from an individual patient and cell type. Box plots show the median (line) and interquartile range (box). The lower and upper whiskers extend to the smallest and largest values within 1.5 times the interquartile range from the first and third quartiles, respectively. Two-sided Wilcoxon rank-sum test with Benjamini–Hochberg correction for multiple comparisons. \**p* < 0.05, \*\**p* < 0.01. **d**–**f** TMT mass spectrometry proteomics analyses is shown for PBMCs from 11 patients with non-SD and SD ($N = 6$ and 5, respectively). Group comparisons were performed with two-sided Welch's *t*-test (unequal variances) between non-SD and SD patients. The *p*-values were adjusted by the permutation-based FDR procedure implemented in Perseus v2.0.7.0 (default setting). **d** Volcano plot showing the differentially expressed proteins in SD versus non-SD patients. Highlighted are interferon-stimulated gene (ISG) products with adjusted *p* < 0.05, log2FC < −1 (2-fold decrease). **e** Heatmap displaying the 54 ISG products that are differentially expressed (adjusted *p* < 0.05) in SD versus non-SD patients. **f** GO over-representation analyses of significantly downregulated proteins [adjusted *p* < 0.05, log2FC < −0.58 (1.5-fold decrease)] in SD vs non-SD patients. In (**b** and **f**), Significant non-redundant gene ontology terms and associated Benjamini–Hochberg adjusted *p*-values are shown. Count = number of differentially expressed genes (DEGs)/proteins associated with each term. GeneRatio = fraction of DEGs in the gene set. Source data are provided as a Source Data file.

unique dengue cohort including a larger number of SD patients than previously analysed in a single study ($N = 30$ SD; $N = 94$ non-SD), as well as sex, age and illness phase matched patients with HW or OW/OB. The latter allows the analysis of the impact of OW/OB, a risk factor for dengue, on immunity to DENV. We show that in early disease, $CD4^+$/$CD8^+$ T and NK-cell profiles linked to activation, proliferation, cytotoxicity, and skin/peripheral tissue homing, discriminate patients that will progress to SD from those that do not. Within non-SD patients, these profiles could discriminate patients with and without warning signs, suggesting that certain warning signs may be immunologically driven. T/NK-cell profiles were distinct between patients with HW, OW and OB, suggesting that BMI impacts immunity to dengue. Furthermore, NK-cell expression of activating/inhibitory receptors and cytotoxic molecules at TP1 also clearly discriminates between SD and non-SD patients. Here, the linear discriminant analyses showed a less clear discrimination of patients based on BMI, suggesting NK-cell responses in dengue may be less impacted by host BMI. For these analyses at TP1, we included SD patients enrolled prior to the development of severe manifestations (SD progressors; days 2, 3 and 4), as well as SD patients who were recruited at the early onset of SD in the ICU (day 5 and one patient at day 4). T/NK-cell profiles were similar in SD progressors and SD patients; hence, these data were pooled together in our analyses. A limitation of these analyses is the difference in the day of illness at TP1 for SD and non-SD patients (average of 4.3 and 2.9 days of illness, respectively). However, analyses of samples taken at days 3–4 only for both patient groups showed similar results, suggesting robustness of our findings. A further limitation of our study is the use of whole PBMCs and K562 target cells to assess NK-cell cytotoxicity, using a previously described approach[35]. Further studies using purified NK-cells are needed to confirm the reduced cytotoxic capacity of NK-cells in SD compared to non-SD patients.

Our data suggest that T/NK-cells responses: (1) could potentially have prognostic value for early stratification of patients more likely to progress to SD; (2) are critical for the early antiviral response and (3) may represent a novel therapeutic target to restore effective antiviral immunity in dengue. Each of these points is discussed below.

While most DENV infections are self-limiting, the high number of symptomatic DENV infections during seasonal epidemics rapidly overwhelms health care systems in dengue-endemic countries. The availability of biomarkers for early identification of patients who will develop SD would improve healthcare effectiveness and patient outcomes. Several studies have proposed candidate biomarkers for SD, but these have shown limited clinical value due to their appearance later in disease or their short half-life[38]. A recent study including >7400 participants proposed a combination of inflammatory (IL-8, CXCL10/IP-10, sTREM-1, and sCD163) and vascular markers (syndecan-1) as potential biomarkers for severe/moderate dengue, all related to excessive activation of macrophages/myeloid cells, the main targets of DENV infection[23]. Excessive macrophage activation is consistent with a

scenario of dysfunctional cytotoxic NK/T-cells leading to impaired clearance of virus-infected cells observed in our study.

Our transcriptional and proteomic analyses suggest that suboptimal type-I IFN responses may underlie defective viral clearance and NK/T-cell impairment in SD. Type-I IFN signalling leads to activation of ISGs and induction of an antiviral state within the cell[39]. Furthermore, type-I IFNs promote NK-cell function, and type-I IFN blockade was shown to inhibit NK-cell responses to DENV-infected cells in vitro[36]. Similarly, type-I IFNs directly support survival, clonal expansion and cytotoxicity of activated T-cells[37,40,41]. Our findings support early DNA microarray studies showing decreased transcription of many genes induced by type-I IFN in patients with dengue shock syndrome compared to those without[42]. A limitation of our scRNA-seq analyses is the use of a probe-based BD Rhapsody approach, which limits analyses to ~500 genes. However, we integrate and confirm our findings of a dampened type-I IFN response in SD at the protein level using a holistic, whole-proteome proteomics approach. A caveat of both scRNA-seq and proteomics analyses is that these analyses were performed on PBMC samples collected on average 1.4 days later for SD compared to non-SD patients (average DOI is 4.4 and 3 for SD and non-SD, respectively). As type-I IFN responses occur early in infection, this difference may bias our data. However, our data aligns with studies in Cambodian patients showing decreased serum IFN-α and IFN-β[43] levels in SD versus non-SD patients, where samples from both groups were taken at day 3 of illness.

Antibody-dependent enhancement (ADE) of infection mediated by pre-existing subneutralizing DENV-specific IgG antibodies is associated with more severe clinical outcomes[12]. This is mediated by binding of DENV IgG Fc portions to Fc-gamma receptors (FcγRs) on the surface of myeloid cells, leading to DENV internalisation and augmented viral replication. Coligation of FcγR and the inhibitory receptor LILRB1 by antibody-opsonised DENV was shown to inhibit FcγR signalling and induction of ISGs[44]. In NK-cells, this interaction may result in reduced ability to mediate antibody-dependent cellular cytotoxicity, further inhibiting virus clearance. Therefore, ADE, decreased type-I IFN signalling, T/NK-cell cytotoxic impairment and decreased survival of DENV-specific T-cells may all represent interconnected events leading to increased viraemia and severe outcomes. Genetic factors (e.g., MICB, NKG2D SNPs) may further contribute to the suboptimal NK/T-cell response in these patients, as they could render individuals less efficient at mounting cytotoxic NK/T-cells. However, the cause-effect relationship between NK/T-cell dysfunction and dengue clinical outcomes remains to be determined.

A defective innate response to DENV infection likely contributes to the increased viraemia observed in SD. This, in turn, may lead to increased antigen presentation and excessive T-cell activation, causing upregulation of T-cell exhaustion markers and increased apoptosis. In contrast, an adequate innate response will support an effective T-cell response as observed in non-SD patients. We report a T-cell phenotype

which is consistent with T-cell dysfunction/exhaustion in SD patients; however, further studies are needed to address in detail the molecular events occurring in these cells. As SD associates with secondary infections, preferential reactivation of pre-existing, low-affinity memory T-cells could also contribute to the altered T-cell response. This is consistent with the higher frequencies of DENV-specific T-cells at TP1 in SD versus non-SD. However, our analysis of the capacity of DENV-specific T-cells to recognise NS3 DENV1–4 peptides in secondary DENV2 infected patients did not show evidence of skewing of the T-cell response towards heterologous DENV serotypes. While using this approach, we cannot exclude that T-cell skewing may be occurring in T-cells targeting specific epitopes; our data suggest it may not be impacting the total T-cell response. Due to an insufficient number of cells at TP1, we were only able to test recognition of all four serotypes at TP2. It therefore remains possible that suboptimal cross-reactive T-cells that are preferentially activated at TP1 may have undergone apoptosis and are not detectable at TP2, this would be in line with their high CD95/Fas expression levels. For these analyses, we utilised 15-mer overlapping peptides which capture T-cell responses restricted to any HLA type. Future studies addressing T-cell responses within specific HLA types associated with protection in dengue are required. Here we investigated DENV-specific T-cell responses in samples from secondary dengue patients with confirmed DENV-2 infection, representing approx. 70% of patients in this cohort. The DENV serotype of their first infection was unknown, and we were therefore unable to stratify patients based on the serotype of their past infection. Our data is consistent with studies in Colombian dengue cohorts showing an NK-cell related signal in SD using bulk RNA-seq[24] or using virus-inclusive scRNA-seq on a cohort of 12 non-SD and 7 SD patients[45]. Another study by the same group demonstrated high PD-L1 expression in SD across different cell types, including myeloid cells[22]. These cells could potentially provide ligands for T/NK-cell expressed PD-1. DENV infection was shown to upregulate HLA class I and non-classical HLA class I molecules, such as HLA-E, which are ligands for NK-cell inhibitory receptors, respectively LILRB1 and NKG2A, expressed in SD[46], suggesting that DENV has evolved strategies to counteract the early NK-cell response. It remains to be determined whether DENV proteins cause upregulation of ligands binding to inhibitory receptors expressed by T-cells to evade the T-cell response.

Lastly, we propose that T/NK-cell impairment may represent a promising therapeutic target for dengue, supporting recent interest in evaluating immune checkpoint blockade in infectious diseases[47]. In acute hepatitis C virus (HCV) infection, PD-1 expression was shown to correlate with CD8$^+$ T-cell exhaustion and PD(L)1 blockade could restore the function of these cells[48]. As for HCV patients, higher PD-1 T-cell expression in SD versus non-SD patients does not reflect the higher activation state of these cells, as expression of other activation markers such as HLA-DR, CD38 and CD69 is similar in T-cells from the two patient groups. Previous studies by us and others reported PD-1 expression in memory DENV-specific CD8$^+$ T-cells in convalescent patients or healthy donors from a dengue hyperendemic region[49–51], although these studies did not address links with SD. Here we show that PD-1/PDL1 blockade can restore the cytotoxic potential of CD8$^+$ T-cells in some patients. Recent work in a symptomatic *Ifnar*$^{-/-}$ dengue mouse model shows a similar increase of PD-1$^+$ CD8$^+$ T-cells upon infection with a non-mouse adapted DENV strain, which leads to plasma leakage and death. PD-1 blockade prior to DENV infection significantly improved mouse survival and rescued CD8$^+$ T-cell numbers, suggesting a role of PD-1 in T-cell apoptosis[52]. These data in humans and mice support the need for further studies evaluating the impact of checkpoint inhibitors or combinations of checkpoint inhibitors on NK/T-cell function in dengue. Studies in non-human primates show that while viraemia is cleared around day 5 from illness onset, similarly to patients, DENV antigens can be detected in tissues until at least day 8[53]. This data supports an important role of cytotoxic NK-cells/DENV-specific T-cells beyond the blood viraemic phase, for clearing reservoirs of DENV-infected cells within tissues.

In summary, our work demonstrates T and NK-cell impairment in SD patients, which precedes the development of SD and is present during the critical phase. Further studies validating these immune signatures as prognostic markers for SD are warranted. We propose immune dysregulation as a novel therapeutic avenue for dengue aimed at restoring the antiviral response in patients.

## Methods

### Ethics approvals

The study protocol, consent and assent forms, and patient information sheets were approved by the ethics committees at the Hospital for Tropical Diseases in Ho Chi Minh City (CS/BND/19/34), the Ministry of Health in Vietnam (24/CN-HĐĐĐ) and the Oxford Tropical Research Ethics Committee (REC) (OxREC reference 36-19:). All samples were handled in line with the Human Tissue Act, and research was conducted under a Health Research Authority REC approval (reference:19/LO/1809).

Written Patient Information sheet and informed consent form (ICF) or assent form (AF) were presented verbally to the patients detailing the exact nature of the study, including what it will involve for the patient, the implications and constraints of the protocol, and any risks involved in taking part. Written Informed Consent/assent was then documented by a patient's dated signature, and the dated signature of the person who obtained the Informed Consent. Illiterate signatories had the ICF/AF read to them in the presence of a witness, who signed to confirm this. Patients who were between 10 and 18 years of age were asked to sign the AF, in which the study purpose and procedures had been explained to them in child-friendly terms. Their parent/guardian was asked to sign the ICF as well, to give permission for their child to participate in the study.

### Dengue patients

After informed consent/assent, hospitalised patients aged 10–30 were recruited into an observational study designed for dengue patients with overweight/obesity and healthy weight at the Hospital for Tropical Diseases, Ho Chi Minh City, Vietnam. All patients had confirmed dengue and ≤72 h of fever, except for a proportion of severe dengue patients who were admitted to the intensive care units (ICUs) up to day 5 of fever. The SD group therefore included patients who progressed to SD during their hospitalisation (SD progressors, SDp: six patients at day 3, two patients at day 2 and two patients at day 4) and patients who were defined as SD at admission/enrolment (SD: three patients at day 4, sixteen patients at day 5 and one patient at day 6—for the latter only samples at TP2 were included in the study). Each patient with overweight/obesity was matched 1:1 to a healthy weight patient by age group (10–16; >16–21; >21–26; >26–30), sex, admission ward (general or ICU), and illness phase—febrile (fever days 1–3) or critical (fever days 4–5). The enrolment criteria were selected to minimise confounders due to ageing and comorbidities and to maximise recruitment of severe cases. The exclusion criteria included hypertension, cardiovascular disease, signs or symptoms of any other acute infectious disease, undernutrition, and pregnancy.

Definition of BMI groups for paediatric patients (10–19 years) was based on the WHO obesity definition using BMI-for-age[54]; for adult patients (20–30 years) BMI status was defined as follows: individuals with overweight/obesity had a BMI ≥ 25 kg/m$^2$, while HW patients had a BMI ≤ 22 kg/m$^2$ but not less than 17 kg/m$^2$. The severity grade classification was recorded for all admitted patients following the WHO 2009 guidelines[55] and based on plasma leakage (grade 0-2)[56]. PBMC isolation and cryopreservation were performed at hospital enrolment (at the febrile or critical phase, days 1–3 and 4–5, respectively) and approx. 3 days later, which largely coincided with discharge (6–9 days of illness).

## Serology

Identification of the DENV serotype of infection and quantification of viraemia (RNAemia) was performed by serotype-specific RT-PCR. The majority of patients were infected with DENV2 serotype ($n = 86$), although some cases of DENV1 ($n = 22$) and DENV4 ($n = 7$) serotypes were also detected. RNAemia was below the detection limit at the time of enrolment in the study for 3 SD and 6 non-SD patients (indicated as N/A; Table S1). Primary/secondary dengue infection was determined based on the ratio of dengue IgM/IgG antibody levels through enzyme-linked immunosorbent assay (ELISA), as described[23]. The majority of patients ($n = 118$) included in the study had a secondary infection; there were five cases of primary infection among the non-SD group. Pro-inflammatory chemokines and cytokines were analysed from plasma samples using a Luminex FLEXMAP 3D® using the Inflammation 20-plex human Procartaplex™ panel. Endothelial, pro-inflammatory, and lipid markers and sMICB were measured by ELISA.

## DENV synthetic peptides

NS3 DENV peptide pools comprised of 15-mer peptides overlapping by 10 amino acids and spanning the sequence of NS3 DENV1−4 (accession numbers: DENV1−MF314188, DENV2−NP056776, DENV3−KY921906, DENV4−KY921909). NS3 peptide pool 1: $N = 40$, pool 2: $N = 40$, pool 3: $N = 42$. Peptide libraries were designed based on the following virus strains: DENV1: 2016_Singapore_DENV-1_NPHL; DENV2: Thailand/16681/84; DENV3: SG(EHI)D3/23167Y15 and DENV4: SG(EHI)D4/09291Y16. All peptides were purchased from Mimotopes (Australia) with >80% purity. Peptides were prepared as described previously[27].

## Ex vivo PBMC surface and intracellular staining

PBMCs were thawed in RPMI 10% FBS, washed twice and seeded at $0.8–1 \times 10^6$ cells/well in a 96-well plate. Cells were stained with a viability dye (Zombie Aqua) for 10 min, washed with PBS 1% BSA and antibodies targeting surface markers were added and incubated for 20 min at 4 °C. After staining, cells were washed and fixed (BD/eBiosciences). After fixation, the cells were washed three times with perm/wash buffer (BD/eBiosciences) and stained intracellularly with a mixture of antibodies targeting cytokines for 30 min at 4 °C. The cells were washed before acquisition using the BD Fortessa X20 flow cytometer. The list of antibodies and reagents is included in Supplementary Table S4. Single-stained compensation controls were included in each experiment for each antibody used.

## T-cell stimulation with DENV peptide pools

PBMCs were rapidly thawed in RPMI 10% FBS, washed twice with PBS 1% BSA and then resuspended in AIM-V 2% AB human serum and rested for 18 h at 37 °C. After resting, cells were seeded at $1 \times 10^6$ cells/well in a 96-well plate in AIM-V 2% FBS. Cells were stimulated with or without (DMSO only−negative control) peptide pools from DENV1-4 (all 1 µg/mL) or with PMA/ionomycin (positive control) for 6 h at 37 °C in the presence of brefeldin A (2 µg/mL). To assess degranulation, anti-CD107a antibody was added at the beginning of stimulation. After stimulation, cells were washed and stained as described above. The list of antibodies and reagents is included in Supplementary Table S4.

## PD-1/PD-L1 blockade

PBMCs were rapidly thawed, washed twice and then resuspended in AIM-V 2% AB human serum and plated at $0.8–1 \times 10^6$ cells/well in a 96-well plate. Blocking antibodies (αPD-1/PD-L1) or isotype controls (IgG4/IgG1) were added at a final concentration of 10 µg/mL. After 2 h pre-incubation, cells were stimulated with NS3 DENV peptide pools (all 1 µg/mL) or with αCD3/CD28 Dynabeads (positive control) for 16 h at 37 °C in the presence of brefeldin A (2 µg/mL), anti-CD107a antibody and blocking antibodies. After incubation, cells were washed and stained as described above.

## NK-cell killing assay

NK-cell killing assays were performed as described previously[35]. Briefly, PBMCs were thawed, washed with RPMI 10% FBS and rested or stimulated overnight with 10 ng/mL IL-12 and 100 ng/mL IL-18 in RPMI 10% FBS at 37 °C. After incubation, K562 cells were added to rested or stimulated PBMCs at an effector-to-target ratio of 10:1 ($1 \times 10^6$ PBMCs: $1 \times 10^6$ K562 cells). Wells containing only PBMCs (no K562 cells) were also included as a negative control. The plate was incubated for 6 h at 37 °C in the presence of anti-CD107a, while monensin (2 µM) and brefeldin A (2 µg/mL) were added 1 h into the assay. After the incubation, PBMCs were subsequently stained as previously described. The list of antibodies is included in Supplementary Table S4.

## BD Rhapsody scRNA-seq library generation and sequencing

PBMCs were stained with CD45-PE for 30 min at room temperature. Cells were washed twice in FACS buffer (PBS 2% FBS), incubated with Fc block solution for 15 min at room temperature, then each sample labelled with a separate oligonucleotide-conjugated Sample Tag (BD Flex Single-Cell Multiplexing Kits A-D) for 60 min at 4 °C. Cells were washed twice in BD Pharmingen Stain Buffer. Twelve samples (3500 cells/sample; $n = 3$ from each group−non-SD HW, non-SD OW/OB, SD HW, SD OW/OB) were pooled and loaded onto one BD Rhapsody Cartridge following the manufacturer's instructions. The remaining 12 samples were pooled and loaded onto a second BD Rhapsody Cartridge. Targeted mRNA and Sample Tag library preparation were performed according to the manufacturer's instructions. Targeted mRNA libraries were generated using the BD Rhapsody Immune Response Panel HS and a custom panel containing an additional 145 genes (Supplementary Table S2). Libraries were quantified using a Qubit Fluorometer with the Qubit dsDNA HS Assay Kit and their fragment distribution analysed using a TapeStation with the High Sensitivity D5000 ScreenTape Assay (Agilent). Sequencing was performed on an Illumina NovaSeq X Plus (PE150).

## BD Rhapsody scRNA-seq analysis

BCL files were converted to FASTQ files using Illumina bcl2fastq. FASTQ files were processed to a cell-by-gene count matrix (one per BD Rhapsody Cartridge) using the BD Rhapsody Sequence Analysis Pipeline (v2.2.1). Downstream analysis was performed in R (v4.4.0-v4.4.2). A Seurat[57] (v5.1.0) object was generated from distribution-based error correction-adjusted molecule counts (two Cartridges combined) and filtered to retain cells with a defined Sample Tag and to remove doublets (cells associated with more than one Sample Tag). Cells with low unique molecular identifier counts (<125) and/or low gene counts (<40) were removed. Count data was analysed following the standard Seurat pipeline. In brief, counts were normalised by library size, scaled, and principal component analysis was performed using the top 200 variable genes. A shared nearest neighbour graph was constructed using the top 20 principal components and Louvain clusters identified (resolution 0.5). UMAP was performed using the top 20 principal components. Reference-based cell annotation was performed using Azimuth (v0.5.0) with the PBMC reference dataset[58]. Cluster markers were identified using the Wilcoxon rank-sum test (Seurat FindAllMarkers function, only.pos=TRUE) and clusters assigned to cell types using expert knowledge. Final cell annotations were determined using a combination of Azimuth results and manual cluster assignments.

Differential gene expression analysis between non-SD and SD (HW and OW/OB patients combined) was performed for all cell types combined, for individual cell types and for total CD4+ and CD8+ T-cells, and total NK-cells using the Wilcoxon rank-sum test (Seurat FindMarkers Function). Significantly differentially expressed genes were defined as those with a Bonferroni-adjusted $p < 0.05$ and an absolute average $\log_2 FC > 0.25$. Over-representation analysis for Gene Ontology (GO) Biological Process terms was performed using cluster Profiler[59] (v4.12.2) for genes significantly downregulated in SD

compared with non-SD (all genes tested for differential expression included as background). Redundant GO terms were removed using the clusterProfiler simplify function. Differential cell type abundance analysis was performed using edgeR (v4.2.1)[60,61]. Namely, negative binomial dispersions were estimated using the estimateDisp function (trend.method = "none"). Quasi-likelihood negative binomial generalised log-linear models were fit to per-sample cluster counts (glmQLFit function, robust=TRUE, abundance.trend=FALSE) and quasi F-tests (glmQLFTest function) employed to test for differential abundance between non-SD and SD (HW and OW/OB patients combined). Differentially abundant clusters were defined as those with an FDR < 0.05 and an absolute log2FC > 1. Per cell type-I IFN signalling module scores were calculated using the Seurat AddModuleScore function (ctrl = 20) with genes from the "type I interferon-mediated signalling pathway" GO term (GO:0060337). Per-sample scores for each cluster or for total PBMCs were generated by averaging (mean) or summing individual cell scores. Statistical comparison between non-SD and SD for each cluster was performed using the Wilcoxon rank-sum test, and p-values were adjusted for multiple comparisons using the Benjamini–Hochberg method.

### Proteomic analysis: sample preparation

PBMC samples from 11 dengue patients (N = 6 non-SD; N = 5 SD) were analysed from the sample set used for scRNA-seq. PBMCs were washed twice with cold PBS, resuspended in RIPA buffer (Thermo Fisher Scientific), and incubated on ice for 30 min, with pipetting every 10 min to ensure complete cell lysis. The lysates were centrifuged at 12,000×g for 10 min, and the supernatants were used for downstream proteomic analysis. The protein concentration in each sample was quantified using an EZQ protein Quantification Kit (Thermo Fisher Scientific).

### TMT labelling and High pH reversed-phase chromatography

Aliquots of 20 μg of each sample were digested with trypsin (1.25 μg trypsin; 37 °C, overnight), labelled with Tandem Mass Tag (TMTpro) sixteen plex reagents according to the manufacturer's protocol (Thermo Fisher Scientific). The labelled samples were pooled and desalted using a SepPak cartridge according to the manufacturer's instructions (Waters, Milford, Massachusetts, USA). Eluate from the SepPak cartridge was evaporated to dryness and resuspended in 20 mM ammonium hydroxide, pH 10, prior to fractionation by high pH reversed-phase chromatography using an Ultimate 3000 liquid chromatography system (Thermo Fisher Scientific). The resulting fractions (concatenated into 15 in total) were evaporated to dryness and resuspended in 1% formic acid before further fractionation and mass spectrometry analysis using an Ultimate 3000 nano-LC system in line with an Orbitrap Fusion Lumos mass spectrometer (Thermo Fischer Scientific) by the University of Bristol Proteomics facility. Full details are included in the Supplementary Methods file.

### Data and statistical analyses

Analysis of flow cytometry data, including spectral compensation, was performed using FlowJo v10.10.0. Quality control was performed using the FlowJo plugin—PeacoQC. Flow cytometry standard files (FSC) were downsampled and subsequently concatenated prior to UMAP and FlowSOM analyses using FlowJo. Statistical analysis and data visualisation were performed using R v4.2.1. Normality of data distribution was assessed by Shapiro–Wilk test. Differences between two patient groups were tested by Mann–Whitney U test. For more than two groups, we used one-way ANOVA (Kruskal–Wallis) followed by Dunn's test for multiple comparisons adjusted using the Benjamini–Hochberg method to control for FDR. Analysis between paired samples was performed using Wilcoxon signed rank test. Correlation analysis was performed using Spearman's rank correlation test with/without correction for multiple comparisons. P-values are indicated as follows: *$p < 0.05$, **$p < 0.01$, ***$p < 0.001$, ****$p < 0.0001$. The LDA combines

clinical information and frequencies/MFIs of markers in the cell subsets, and it was calculated using the MASS R package. Data analyses were performed by grouping data by specific time points (TP1 and TP2) as well as by day of fever—we report the most informative analysis.

Proteomics data analysis: The raw data files were processed and quantified using Proteome Discoverer software v2.4 (Thermo Fisher Scientific) and searched against the UniProt Human database (downloaded January 2024: 83,095 entries) and a bespoke dengue virus database using the SEQUEST HT algorithm. Peptide precursor mass tolerance was set at 10 ppm, and MS/MS tolerance was set at 0.6 Da. Search criteria included oxidation of methionine (+15.995 Da), acetylation of the protein N-terminus (+42.011 Da) and methionine loss plus acetylation of the protein N-terminus (−89.03 Da) as variable modifications and carbamidomethylation of cysteine (+57.0214 Da) and the addition of the TMTpro mass tag (+304.207 Da) to peptide N-termini and lysine as fixed modifications. Searches were performed with full tryptic digestion, and a maximum of two missed cleavages was allowed. The reverse database search option was enabled, and all data was filtered to satisfy false discovery rate (FDR) of 5%. To identify proteins that were significantly differentially regulated between conditions, the data were analysed using Perseus v2.1.3[62] with statistical analysis using a Welch's t-test and a permutation-based FDR. Visualisation of the data was done using RStudio v4.4.2 with ggplot2[63] and pheatmap packages[64], and software in the Bioconductor repository[65] including EnhancedVolcano[66]. The ISG products used for comparison were obtained from a previous study[67]. Gene ontology overrepresentation analysis was done in RStudio using ClusterProfiler v3.20[68], with the background list set to contain all differentially expressed proteins identified.

### Sex as a biological variable

Data are derived from both females and males.

### Reporting summary

Further information on research design is available in the Nature Portfolio Reporting Summary linked to this article.

## Data availability

Flow cytometry data have been deposited in the Flow Repository database under accession codes: FR-FCM-Z8DN; FR-FCM-Z8E3; FR-FCM-Z8EU; FR-FCM-Z8EV and FR-FCM-Z8EG. ScRNA-seq data have been deposited at the Gene Expression Omnibus database under accession number GSE280483. The mass spectrometry proteomics data have been deposited in the ProteomeXchange Consortium via the PRIDE partner repository with the dataset identifier PXD061694. http://www.ebi.ac.uk/pride. Project accession: PXD061694. Source data are provided with this paper.

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

## Acknowledgements

The authors wish to acknowledge the assistance of Dr Andrew Herman, Helen Rice, Poppy Miller, Celyn Dugdale, Marieangela Wilson, Phil Lewis and the University of Bristol Faculty of Biomedical Sciences Flow Cytometry and Proteomics facilities. We are grateful to all patients and their families for participating in this study. This study was supported by the Academy of Medical Sciences and the Springboard Award scheme funders: the Wellcome Trust, the Government Department of Business, Energy and Industrial Strategy and the British Heart Foundation and Diabetes UK [SBF007\100173]; the Royal Society (RGS\R1\221078). M.S. was supported by the Elizabeth Blackwell Institute for Health Research, University of Bristol, with funding from the University's alumni and friends (TRACK award to L.R.). P.K. and L.G. are supported by the Wellcome Trust (222426/Z/21/Z). R.F.H. is supported by the Indonesia Endowment Fund for Education Agency (LPDP), Ministry of Finance, Republic of Indonesia.

## Author contributions

Conceptualisation, L.R. and S.Y.; Methodology, L.R., M.G., M.S., C.L., P.K., A.D.D., K.J.H.; Validation, M.G., L.R., M.S., D.D., M.N.; Formal Analysis, M.G., M.S., D.D., L.C.G., R.F.H., N.R., M.N., V.L.N., A.D.D.; Investigation, M.G., M.S., D.D., R.F.H., E.J., N.R., N.M.N., V.T.T., C.Q.H., C.T.X.N., T.T.H.D., D.T.L.H., T.T.C., K.J.H.; Resources, L.R., S.Y.; Data Curation, M.G., L.C.G., R.F.H.; Writing—Original Draft, M.G. and L.R., Writing—Review & Editing, M.G., L.R., S.Y., P.K., L.C.G., N.M.N., H.T.M.V., C.Q.H., A.D.D.; Visualisation, M.G., L.R., L.C.G., M.N.; Supervision, L.R.; Project Administration, L.R.; Funding Acquisition, L.R. and S.Y.

## Competing interests

The authors declare no competing interests.
