## [Peer Review file · Nature Communications]

Early NK-cell and T-cell dysfunction marks progression to severe dengue in patients with obesity and healthy weight

Corresponding Author: Dr Laura Rivino

Version 0:

Reviewer comments:

Reviewer #1

(Remarks to the Author)

In the study "Early NK-cell and T-cell dysfunction marks progression to severe dengue in patients with obesity and healthy weight", Gregorova and colleagues aimed to explore the phenotypic, functional, and transcriptional dysfunctions of T and NK cells in patients with severe dengue (SD) or non-severe dengue (non-SD) during the disease course using flow cytometry and scRNA-seq. They found that CD4⁺/CD8⁺ T cells and NK cells exhibited increased expression of coinhibitory receptors, reduced cytotoxic capacity, and downregulated type-I interferon signaling in SD compared to non-SD patients. While this finding is potentially significant, there are several areas where the study could be improved. Below are my suggestions, which I hope will enhance the quality of the study. Specific comments are as follows:

Point 1. The authors used "T1" and "T2" to indicate different stages of the disease course, which may cause confusion, as these terms are often associated with "treatment" in medical literature. The authors did not mention whether different patient groups received distinct treatments, and it is important to consider that different therapeutic agents could influence the outcome. This should be clarified in the manuscript.

Point 2. The authors performed extensive flow cytometry experiments and FlowSOM analysis. It would be important to confirm whether they accounted for potential issues such as differences in cell counts, batch effects across multiple samples, and fluorescence spectral overlap. The manuscript seemingly mentioned the use of two flow cytometry panels, but the specific antibodies used were not listed. Additionally, since flow cytometry typically requires cell stimulation with activation cocktails (PMA/ionomycin/BFA) before measuring inflammatory factors, the authors should clarify whether such stimulation was applied and discuss the potential impact on their results.

Point 3. In the scRNA-seq analysis, the authors did not account for the proportion of mitochondrial genes during quality control. Additionally, for differential gene expression analysis, the threshold for log₂ fold-change (`logfc.threshold=0`) may be too permissive. This cutoff might fail to capture biologically significant changes. A more stringent threshold, such as $|\text{Log}_2\text{FC}| > 0.5$ or at least > 0.25 , should be applied to better reflect the transcriptomic landscape.

Point 4. The order in which the results were described in the manuscript did not correspond with the order of the figures. This lack of alignment is scientifically illogical and could hinder the reader's understanding. I recommend reorganizing the text to match the figure sequence for better readability.

Point 5. In Figure 4, the authors described T cells in SD patients as having increased cytokine production along with high levels of PD-1 and CD95. The authors should verify the cell apoptosis of T cells at T1 and T2 stages in both SD and non-SD patients, using *in vivo* and *in vitro* experiments to further support their findings.

Point 6. The authors reported impaired type-I interferon signaling in SD based on the scRNA-seq analysis but do not conduct further phenotypic or functional experiments to validate this observation. It would be helpful to include additional experiments to confirm this finding.

Point 7. The authors did not specify catalog or clone numbers for the antibodies or reagents used in the study, particularly for PD-1/PD-L1 blocking antibodies.

Point 8. The DENV2-specific T-cell function analysis is unclear. After NS3 DENV2 peptide stimulation, the authors reported that DENV2-specific T cells were predominantly monofunctional. It is unclear what is meant by "DENV2-specific T cells" here—does this refer to all stimulated T cells? Is there literature supporting that T cells after DENV2 peptide stimulation are truly DENV2-specific? Are these cells related to Ki-67⁺CLA⁺ bona fide DENV-specific T cells? Furthermore, how does the author define T-cell function using Boolean gating? Does the presence of IL-2 in these cells indicate that they only express IL-2, or could they be producing other cytokines as well?

Point 9. Did the authors consider or apply corrections for multiple testing in their statistical analyses?

Point 10: In the NK-cell killing assay, PBMCs were co-cultured with K562 cells. However, this experiment does not directly demonstrate the effect of K562 stimulation on NK cells. To strengthen the results, the authors should co-culture sorted NK cells with K562 cells and perform subsequent experiments.

Point 11: The authors used classic marker-based tests to explore the association between phenotypic and functional impairment of CD4+ T, CD8+ T, and NK cells with SD in patients enrolled at the onset of SD. While partial validation experiments were performed, many of the descriptions of phenotypic outcomes appear overstated. These claims should be reconsidered or clarified.

Point 12: The manuscript requires in-depth editing to address grammatical and formatting issues.

Reviewer #2

(Remarks to the Author)

Dear authors,

The manuscript "Early NK-cell and T-cell dysfunction marks progression to severe dengue in patients with obesity and healthy weight" was reviewed. The authors evaluated T and NK cells function from PBMC of dengue infected patients with different severity outcomes using mainly multi-color flow cytometry, ex-vivo stimulation with NS3 overlapping peptides with some additional plasma mediators, metabolic and scRNASeq. The strengths of the study is the high number of samples in the cohort and functional analysis including NS3 overlapping peptide stimulation. However, the sample collection timing differences between samples and confounding factors should be considered. The authors conclude that SD has dysfunctional NK and T cells potentially leading to severe disease but the data is not all conclusive. The presence of higher PD-1 T cells is evident but in functional study, NS-3 specific CD8 T cells in SD has higher CD107a suggesting they might be functioning. Therefore, interpretation of dysfunctional T cells based on PD-1 expression should be taken with caution. scRNASeq data is presented in the last figure but with limited result, the authors should be able to dive more in-depth in the analysis to provide more in-depth understanding of potential molecular mechanisms of the findings found by FACS and DV peptide stimulation.

Major comments

1. Sample collection timing and potential confounding factors. Sample collection timepoints T1 span several days, as immune response to DV is known to be highly dynamic during febrile phase and at critical phase. It is not clear from table S1 which day in each donor is considered as T1 and T2. It is also not clear whether T1 is during febrile phase in all donors or also include day after fever subsided. There is one sentence mentioned that SD include both day 3 (likely during febrile phase) and day 5 (in critical care, likely during critical phase). It is also not clear which clinical stage of T2 is. It was mentioned, T2 is approximately 3 days after T1, do patient still has fever at this point? Is it the same in all patients?

It might be clearer to analyze the result day by day in relation to the day of fever subsided comparing between disease severity or at least consider day of sampling as potential confounder and perform statistical analysis accordingly. This is of concern because SD T1 sample collection seems to be later (majority of samples were collected on day 5 vs Non SD, majority were collected on day 2-3 after symptom onset) (Sup table 1). This might affect the results particularly IFN I is known to upregulate early and go down afterward. As OW/OB seems to have more severe disease, how could we tease out the effect of body weight vs severity on immune profiles? Other additional potential confounders should be included in table S1 eg. DV serotype and viremia level at T1 and T2, body weight in different severity groups. Proper statistical test should be performed to evaluate any confounding effects of these variables.

2. Fig 2h. If the author could annotate clusters (using combined data) before describing details, it would be easier to understand the biological significance of the findings.

3. Line 205 "HLADR+CD38+CD8+ T correlate with plasma levels of leptin suggesting that obesity could have impact on CD8 T cell activation"... but Fig 3 F, more directly, show that OW/OB not correlate with CD8 activation. Please be careful not to over interpret the result.

4. NS3- specific T cells response

a. It would be important to know previously infecting serotype and analyse the data separating previous vs current infecting serotype. Test such as PRNT could be performed to identify previously infecting serotype.

b. As antigen-specific T cell response depends on HLA, have the authors adjusted of donors' HLA?

c. Fig 4c high variability is observed.

d. It would be interesting to further investigate which epitope of NS3 are dominant.

e. It would be of great interest if scRNASeq could be performed after NS3 stimulation.

f. Line 253 "SD patients also display increased frequencies of degranulating CD107a+". This could suggest that NS-3 specific CD8 T cells in SD, even though they express high PD-1 and inhibitory co-receptors, they are functional. Consistent with previous report in COVID that PD-1-expressing SARS-CoV2-specific CD8+ T cells are not exhausted but functional. (<https://www.sciencedirect.com/science/article/pii/S1074761320305094>)

5. Fig 5k, data is highly variable, only 2 in 9 donors showed increase CD8+GzB+Perf+ upon PD-1 blockade, please perform statistical test including all samples. It is difficult to draw conclusion from the data presented.

6. Figure 7. scRNASeq

- The limitation of probe-based BD rhapsody without TCR/ BCR data should be discussed

- A lot more in-depth analysis of scRNASeq data can be done and would potentially provide more in-depth understanding of

the molecular mechanism of T and NK cell response as well as cell-cell communication with other immune cells within PBMC. DEG in other cell clusters should also be described. The authors should not only limit their analysis to differential cluster abundance and GO term in all clusters combined.

7. The authors claim that IFN- γ is impaired in SD, based on scRNASeq data. As in comment 1, it is important to align timepoint of sample collection between groups for fair comparison. Further, protein level confirmation such as serum level of IFN- γ and intracellular cytokine staining of IFN- γ would be crucial.

Minor comments

1. Please show FACS gating strategy with FMO for main Fig 2 and 3 in supplementary data.
2. Ki67+CLA CD4+ T cells probably not enough to be called "Bona fide DENV-specific T cells" as antigen specificity is not directly shown. (line 148)
3. All data should be deposited and allow access to audience.

Reviewer #3

(Remarks to the Author)

This paper by Gregorova et al describes early NK-cell and T cell dysfunction in patients which develop to severe dengue. The paper is build around an in depth phenotypic analysis of dengue patients, and compares phenotypes at different timepoints in the disease, between non-severe and severe dengue and between overweight/obese patients and patients with healthy weight. With all these comparisons, the paper feels diluted in its messages and lacks clear conclusions. It also makes the figures sometimes difficult to understand.

Major concerns:

1. Dengue disease is very dynamic, and changes in immune cell composition occur rapidly. Patients are at risk to develop severe disease in a small window after onset of symptoms, usually 3-6 days post onset of symptoms (WHO guidelines, 2009). Therefore, including patients up to 5 days of fever in the SD group and not in the non-SD group will introduce a large variability in the immune cell phenotypes and functions. Looking at Table S1, only 13 non-SD were included at day 4 and none later (14% of the non-SD patients), while in the SD group 21 patients were enrolled at day 4 or later (72% of the SD patients).

2. A major concern is the interpretation of the data and the conclusion that the authors have identified prognostic signatures before the onset of severe dengue disease. However, the critical phase is between 3-6 days of illness. Can the authors confirm their most important findings when selecting only patients before the critical phase? This analysis is also needed if the authors want to conclude on NK and T cell dysfunction in patients progressing to severe dengue, as stated in the title, abstract and discussion. In case that is not possible due to the low number of patients in the severe disease group, the title, and interpretation should be rephrased, refraining from concluding these signatures are prognostic markers for SD.

3. Similarly, previous immune history has a major impact on immune cell phenotypes and function. Line 702 mentions 5 cases of primary infection in the non-SD group, could the authors confirm the most important findings when excluding these primary infections?

4. the gene expression data are obtained from a gene expression analysis of a targeted set of genes. This experiment has other important limitations, such as the pooling of 12 individuals per cassette. Both of these important informations should be mentioned in the results section before describing these results.

5. Samples were pooled on only 2 BD Rhapsody Cartridges, 3500 cells each, which is over the maximum capacity stated by the company (around 40000, see e.g. <https://www.bdbiosciences.com/content/dam/bdb/marketing-documents/products-pdf-folder/instruments/bd-rhapsody-express-single-cell-analysis-system/BD-Rhapsody-Single-Cell-Analysis-System-Brochure.pdf>). Thus it is crucial to add tabs and QC plots to show basic statistics of the scRNA-seq analysis, such as the number of cells before and after filtering steps, the number of identified doublets, the distributions of the number of RNA molecules per cell, the number of different genes per cell, the rate of mitochondrial genes and the cell cycle stage of the cells for quality control purposes.

6. The differential gene expression analysis was performed using the non-parametric Wilcoxon Rank Sum test, which is the default method used in Seurat (originally for performance reasons). While Wilcoxon has been indicated by a study by Jingyi Jessica Li group (Li et al., 2022: 10.1186/s13059-022-02648-4) as the best method for controlling FDR, the results of this study remain controversial in the community (see e.g. https://www.reddit.com/r/bioinformatics/comments/tg8v7z/paper_exaggerated_false_positives_by_popular/ and https://www.reddit.com/r/bioinformatics/comments/161p3xx/wilcoxon_rank_sun_test_as_alternative_to_deseq2/). In addition, several other comparative studies (e.g. Miao and Zhang 2016: 10.1007/s40484-016-0089-7, Sonesson and Robinson, 2018: 10.1038/nmeth.4612, Li et al., 2022: 10.1371/journal.pone.0264246) rather indicate the use of methods such as DeSeq2, which considers sample technical heterogeneity, as preferred. Therefore, could the authors reanalyse the data with a method which considers sample technical heterogeneity, such as DeSeq2, as e.g. shown in the Seurat vignette (https://satijalab.org/seurat/articles/de_vignette).

Minor concerns:

Figure 1a should include immunophenotyping below single-cell RNA sequencing, as most data (Fig1-6) are actually generated by flow cytometry.

Figure 2i and 3l, this reviewer does not understand what is shown in the top part of the figure (%events on y axis) or what the arrows indicate? maybe a more clear description or figure can help.

Line 430-430: comparisons throughout the paper are made between SD and non-SD, could the authors expand on the results supporting this statement?

Line 677 mentions "a proportion of severe patients were admitted up to day 5 of fever". This is also shown in Figure 1a. It should be added how many patients this is, both in the methods text and the figure (data can be found in table S1)

Reviewer #4

(Remarks to the Author)

NCOMMS-24-65884 (Gregorova et al)

In this study Gregorova et al have comprehensively characterized the phenotypic, functional and transcriptional analyses of T and NK-cells in a cohort DENV-infected subjects during the course of acute infection. This study offers a unique insight into T/NK cell biology during DENV-infection in the context of weight/obesity stratification. The study is well-designed and well-presented and an important contribution to the field. Pending the authors addressing several minor issues satisfactorily, I would recommend this paper for publication:

1. In general, it is unclear how the breakdown of HW versus OW/OB stratifies between those under 18 and those over 18 (i.e. children versus adults). I'm assuming the OW/OB group falls more within the adult (over 18) age-group. Hence, while still very interesting, is the phenomenon presented more of a children-versus-adult finding or age-related? Can age be taken account of using a multi-variable confounder analysis?

2. Figure 1B: What is the denominator for the presented frequencies (percentages)? Total lymphocytes? All PBMC? Since percentages are essentially a zero-sum game, increases in one or more phenotypes are associated with similar decreases in others.

3. Lines 133-134. It is stated that "T/NK-cell profiles are more strongly impacted by clinical outcomes than BMI". This would suggest that T/NK cell profiles are reactive to the infection outcome than driving it perhaps. Is this what the authors are really trying to say here? This is important as it gets at the question of cause and effect, which is the core question in such studies. It certainly appears from subsequent data that NK cells are more likely the drivers (if at all) rather T cells.

4. Lines 147-148: Using Ki-67 and CLA co-expression as a surrogate for DENV-specific T cells is fine but using the term bona fide DENV-specific T cells is a stretch without having an example of confirmation of the specificity.

5. Lines 161-162: The terms protective role and detrimental role are only suggestive. Those phenotypes could be resultant of the outcome, and not necessarily driving it.

6. Figure 2 panel H is a little confusing: It looks like the total number of subjects T1 = 42 while the total at T2 is 84. Were any of these the same subjects? I'm assuming that they were. It is understandable with such studies that getting matched samples for subjects is difficult. For the SD subjects at T1 the sample numbers are lower (n=11) than the others. Is this why the dot-plot appears less dense? Could it be that some clusters have decreased, rather the identified clusters having simply increased?

7. The directly comparative analyses of CD4+ and CD8+ T cells shown in Figures 2 and 3 is greatly appreciated and clearly highlights the important features and differences between the two lineages and between T1 and T2 sampling times.

8. Could the functional data in Figure 4F be presented more like the phenotyping data in Figure 5E? It looks like the Spice application (or something similar) was used to generate the figure. The functional data may be more understandable in this format and is buried in Figure S3. I do understand that figure space may be limited though. Was any kind of sub-analysis of the functional studies performed to affirm using Ki-67 and CLA co-expression as a surrogate for DENV-specific T cells?

9. Lines 257-259, and a more general critique: Since the T cell functional response increases disproportionately in SD compared to non-SD patients is it not simply possible that early and more rapid viral replication and hence antigen load is driving this T cell response? The SD outcome phenotype may be "set" earlier and the T cell response is reactive to this. Better innate control leads to development of an appropriately developed T cell response phenotype while an "overwhelmed" innate response results in driving a dysfunctional and exhausted T cell response.

10. Lines 321-323: Since the PD-1 data supports glycolysis as the preferred metabolic usage pathway in OB/OW versus HW cells, does the corollary hold true: i.e. in HW patients T cells are more prone to using anabolic metabolism? The methodology used here allows for a crude analysis of metabolic pathway usage and perhaps a more in-depth metabolic

pathway analysis is required to answer this. In general, the observation here is in line with PD-1+ cells being metabolically more quiescent and non-proliferative which makes sense.

11. Figure 6: The NK cell data is well presented, profound and of note. The main question I would have, is whether the observed NK cell deficit of function could be specific to the response to DENV-infection, or whether it is a generalized effect. i.e. if one were to analyze influenza or other viral infections (ssRNA viruses in particular) in the same way, would the same observation of NK cell impairment be true?

12. It appears that overall innate sensing dictates the potential pathway to SD versus non-SD and that and genetic and behavioral influences such as in this case BMI and metabolism play a role in steering the outcome positively or negatively. One question which remains is relationship of the outcome T cell response and its effect on clinical outcome. This is still an outstanding question. It does not impact the quality of the study, but the authors should be clear that this is the case.

13. Lines 500-502: It is difficult to envision a scenario in which checkpoint inhibition could be applied in a systematic and effective way to modify DENV outcomes as a treatment strategy. How would one identify patients to treat when it seems like the effect is already set by the time patients would come to a clinic? Symptomatic treatment of plasma leakage still appears the best strategy.

14. While the scRNA-seq analysis is well performed it is unclear exactly what it adds to the study. For panel C. Figure 7, does this refer to all cells or is it "gated" or derived from a certain subset of cells within the PBMC? Perhaps Figure S7 shows this better as it focuses on CD8+ and NK cells, which were defined as important by the flow cytometry data. Since this scRNA-seq method (BD Rhapsody) is heavily dependent/biased on the pre-selected target genes it is perhaps not as powerful as drop-seq (such as 10X genomics) based technologies, which are agnostic and unsupervised.

Version 1:

Reviewer comments:

Reviewer #1

(Remarks to the Author)

I appreciate the authors' efforts in addressing most of the comments and providing additional valuable data. However, I still have concerns about the response to Point 10 regarding the use of PBMCs rather than purified NK cells. PBMCs contain various immune cell types, which may confound the interpretation of NK cell-specific effects. The K562 cell line might also influence other immune cells, indirectly affecting NK cell function. Similarly, other related NK-cell killing assays should also consider this issue. Additionally, the authors cite a 2004 study, which may be outdated given recent advancements in NK cell research. I recommend considering more recent literature and adopting purified NK cell assays to provide more direct evidence of NK cell function.

Reviewer #2

(Remarks to the Author)

The authors addressed most of my concerns and performed additional experiments when necessary. Below, please find the list of minor comments by item listed in the initial revision.

1. Please explicitly state the limitation of different days (TP1) in SD and non-SD. Please also include the rebuttal analysis only day 3-4 (TP1) to supplementary.

Please add the viral load data of each donor on TP1 in supplementary table even though TP1 might not be peak viral load, but it is the time point in which PBMC were obtained for immunological analysis.

4a. Please add the explanation and limitation of identifying previously infecting serotype in the manuscript

4a. Please add the study across HLA type to future perspective

4f. It would be helpful to add discussion of dysfunctional/exhaustion and the need of further studies to better address the molecular mechanism of these cells.

6. Cell-cell interaction could be included in the manuscript sup Fig.

Reviewer #3

(Remarks to the Author)

I commend the authors for their thorough and thoughtful responses to the reviewers and the addition of new proteomic data to strengthen the conclusions. Many more details have been given with respect to data acquisition and analysis, which substantially strengthens the results.

I still have the following remarks:

- As secondary infection is an important risk factor, and therefore, an important confounder, I would like to see the analysis

with only secondary cases included in the manuscript

- I am still not convinced one can use the terminology "progression". As the authors mention themselves, "critical phase is day 4/5-7 from onset of fever. LSA includes 11 patients: 4 were recruited prior to SD, seven classified as SD at admission. So TP1 (less or equal than 5 days) overlaps with onset of severity (day4-5). Only when including individuals prior to day 4 after onset of symptoms), one could claim to have a prognostic immune signature.

- lines 275-283: Some regions of NS3 are conserved between different DENV serotypes, so the peptide pools will contain peptides shared between DENV2 and DENV 1-4. I am not sure from these analysis how one can confirm/dismiss skewing of the immune response to the other serotypes?

Reviewer #4

(Remarks to the Author)

I am happy that the authors have responded sufficiently to all comments.

POINT BY POINT RESPONSE TO THE REVIEWERS

Reviewer #1

In the study "Early NK-cell and T-cell dysfunction marks progression to severe dengue in patients with obesity and healthy weight", Gregorova and colleagues aimed to explore the phenotypic, functional, and transcriptional dysfunctions of T and NK cells in patients with severe dengue (SD) or non-severe dengue (non-SD) during the disease course using flow cytometry and scRNA-seq. They found that CD4+/CD8+ T cells and NK cells exhibited increased expression of coinhibitory receptors, reduced cytotoxic capacity, and downregulated type-I interferon signaling in SD compared to non-SD patients. While this finding is potentially significant, there are several areas where the study could be improved. Below are my suggestions, which I hope will enhance the quality of the study. Specific comments are as follows:

We thank the reviewer for acknowledging the significance of our findings, and for their useful suggestions which have helped to improve the quality of our manuscript.

Point 1. The authors used "T1" and "T2" to indicate different stages of the disease course, which may cause confusion, as these terms are often associated with "treatment" in medical literature. The authors did not mention whether different patient groups received distinct treatments, and it is important to consider that different therapeutic agents could influence the outcome. This should be clarified in the manuscript.

We thank the reviewer for this comment and apologize for the confusion created by the terms "T1" and "T2". Patients included in this study were recruited as part of an observational study for dengue and none of the patients received treatment as no approved treatment exists for dengue. To avoid confusion we have changed the abbreviation of timepoint from "T" to "TP" (TP1 and TP2) and clarified that this is a timepoint of disease (line 109 and throughout the whole manuscript; Fig. 1 legend, line 641)

Point 2. The authors performed extensive flow cytometry experiments and FlowSOM analysis. It would be important to confirm whether they accounted for potential issues such as differences in cell counts, batch effects across multiple samples, and fluorescence spectral overlap. The manuscript seemingly mentioned the use of two flow cytometry panels, but the specific antibodies used were not listed. Additionally, since flow cytometry typically requires cell stimulation with activation cocktails (PMA/ionomycin/BFA) before measuring inflammatory factors, the authors should clarify whether such stimulation was applied and discuss the potential impact on their results.

Thank you for this question. Our flow cytometry experiments are performed using the following rigorous conditions which are routinely used in our laboratory. For all experiments we plated an equal number of cells for each patient sample and for each condition ($0.8-1 \times 10^6$ cells; lines 879; 892; 901). All antibodies used are listed in the Key resources table provided at submission of the manuscript (lines 886-897; 916-917). Antibodies were used at an optimal concentration which is determined by prior titration of each antibody in PBMCs. Fluorescence minus zero (FMO) controls are used to accurately discriminate positive and negative populations for markers which are dimly expressed or have a continuous staining pattern. For every experiment we included single-stained compensation controls

for each antibody, using either compensation beads and/or PBMCs (now included in lines 886-887). As explained in line 1000, prior to running FlowSOM and UMAP analyses we perform the Peak Extraction and Cleaning Oriented Quality Control (PeacoQC) analyses using the Flowjo plugin PeacoQC. This algorithm performs a quality control by checking for regions of irregularity in the flow data. Data is then downsampled to include equal number of cells from each sample and subsequently concatenated. For clarity we have now included more information for our analyses pipeline (lines 1000-1002).

All immunophenotyping is performed directly *ex vivo*, without stimulation of cells to ensure there is minimum manipulation of cells and we are capturing the state of these cells during disease (**Figs 1-3; Fig. 5a-g; Fig. 6a-k**). For the functional assays stimulation is required to test the functional capacity of cells (cytokine-producing and degranulation/cytotoxic potential). For these experiments the stimuli and conditions used are indicated in each figure legend and in the methods section, as summarised below. For major clarity we have now also added any missing details in the main text of the results section (lines 265-267).

- **T-cell stimulation:** **Fig. 4** legend (lines 714-716) and methods (lines 892-894) and results section (lines 265-267).
- **PD1/PDL1 blockade:** Fig. 5 legend (lines 762-764), methods (lines 902-904) and results (lines 369-373).
- **NK-cell killing assays:** Fig. 6 legend (lines 793-796), methods (lines 909-913) and results sections (lines 427-432).

Point 3. In the scRNA-seq analysis, the authors did not account for the proportion of mitochondrial genes during quality control. Additionally, for differential gene expression analysis, the threshold for log2 fold-change ($\text{logfc.threshold}=0$) may be too permissive. This cutoff might fail to capture biologically significant changes. A more stringent threshold, such as $|\text{Log2FC}| > 0.5$ or at least > 0.25 , should be applied to better reflect the transcriptomic landscape.

ScRNA-seq was performed by BD Rhapsody using a targeted gene panel - Immune Response Panel HS (approx. 300 genes) and a custom panel containing an additional 145 genes. Since no mitochondrial genes are present in the panel, we cannot use mitochondrial percentage as a QC metric for this data. The choice of using BD Rhapsody for these analyses was dictated by complex biosafety issues as UK regulations require us to perform these experiments in a containment level 3 (CL-3) laboratory. We have added a statement in the results and discussion sections acknowledging the limitations of using a probe-based method for scRNA-seq (lines 469-471; 539-540)

We have now performed additional analysis and implemented the additional filtering suggested by this reviewer. With filtering for absolute $\text{avg_log2FC} > 0.25$, the differentially expressed gene lists for individual cell types remain the same, except for one gene which is lost in the monocyte population. For the analysis of all cell types combined, the number of significant genes reduces from 144 to 115 with filtering for absolute $\text{avg_log2FC} > 0.25$; genes involved in type-I IFN signalling remain significantly downregulated in severe dengue patients. We have substituted the new table of genes obtained with these new more stringent analyses in **Supplementary Table S3**.

Point 4. The order in which the results were described in the manuscript did not correspond with the order of the figures. This lack of alignment is scientifically illogical and could hinder the reader's understanding. I recommend reorganizing the text to match the figure sequence for better readability.

Thank you for this comment and recommendation and we apologise for these discrepancies. We have now reordered the text relative to Fig. 2 and Fig. 3 to reflect the order of the subfigures (**Fig. 2: lines 175-186; Fig. 3: lines 226-231**).

Point 5. In Figure 4, the authors described T cells in SD patients as having increased cytokine production along with high levels of PD-1 and CD95. The authors should verify the cell apoptosis of T cells at T1 and T2 stages in both SD and non-SD patients, using in vivo and in vitro experiments to further support their findings.

Based on data from **Fig. 4h** we suggest that dengue NS3-specific PD1⁺ T-cells may be undergoing increased apoptosis in SD due to their significantly higher expression of CD95 compared to their PD-1⁻ counterparts. We agree that it would be interesting to further explore and validate these findings. However, in vivo experiments remain challenging due to the lack of an animal model for dengue that is suitable for immunological analyses, as dengue mouse models which recapitulate plasma leakage and severe dengue are partially immunocompromised, type-I IFN α -/-mice where T/NK-cells have functional defects (e.g., Pei Xuan Lee et al. JEM 2002).

More extensive ex vivo experiments in dengue-specific T-cells of patients would require samples from a new dengue cohort which is beyond the scope of the current study. We have toned down our statement at lines 310-311 “*suggesting these cells may be undergoing cell death*”.

Point 6. The authors reported impaired type-I interferon signaling in SD based on the scRNA-seq analysis but do not conduct further phenotypic or functional experiments to validate this observation. It would be helpful to include additional experiments to confirm this finding.

We thank the reviewer for this comment. We have now performed additional experiments to validate our findings of the decreased type-I IFN signalling at the protein level, using two different methods: TMT mass spectrometry-based proteomics and Luminex analyses. These additional data which are described below and included in the revised manuscript, expand and validate our scRNA-seq findings, confirming that SD patients display decreased type-I IFN responses compared to non-SD patients.

1. Proteomics was performed on PBMC samples that were included in the scRNA-seq analyses and for which we had an additional PBMC vial available. Data is included in results (lines 471-488 and copied below), **Fig. 7d-e** and Fig.7 legend (lines 808-818)

“A limitation of our scRNA-seq data is the use of a probe-based approach which limited our analyses to ~500 genes and precluded a comprehensive characterization of the gene expression landscape. To expand our analyses and validate at the protein level the dampened type-I IFN signalling observed by scRNA-seq, we performed a quantitative Tandem Mass Tag (TMT) mass-spectrometry based proteomics analyses of PBMC samples from 11 dengue patients at TP1 (N=6 non-SD; N=5 SD). This holistic whole-proteome analyses identified >6700 proteins of which 1655 are differentially expressed between SD and non-SD patients (adjusted p<0.05), with 280 displaying a >1.5-fold downregulation in SD (log₂FC<-0.585; **Fig.7d**; **Supplementary Table S4**). Screening of the identified proteins against a list of 350 proteins encoded by known interferon stimulated genes (ISGs) identified 54 ISGs that are significantly dysregulated (adjusted p<0.05, listed in **Fig.7e**), 27 with a 1.5-fold decrease (log₂FC<-0.585) and 12 with a 2-fold decrease (log₂FC<-1, highlighted in **Fig.7d**). GO ORA of differentially expressed proteins revealed a downregulation of the innate immune response in SD as well as other biological processes such as cell migration and adhesion (**Fig.7f**).

In summary, using transcriptional and proteomics approaches we demonstrate decreased expression of genes and proteins involved in type-I IFN signalling in SD”.

2. **Luminex:** We investigated the levels of IFN- α and IFN- β using the Thermo Fisher human IFN- α and IFN- β Luminex simplex kits in plasma samples from 35 dengue patients (N=18 SD; N=17 non-SD) at the two timepoints (TP1 and TP2). Samples from 24 of these patients had been analysed by scRNA-seq (see Fig 7 a-c). IFN- β levels were below the detection limit for all samples, in line with the typically low concentrations of this cytokine in circulation (data not shown). IFN- α levels were higher in the plasma of non-SD compared to SD patients at TP1, and decreased from TP1 to TP2, with levels largely below the detection limit in TP2 (latter data is not shown; **Supplementary Fig. S10a**). For the 24 patients for which we had matched scRNA-seq and Luminex data, the plasma levels of IFN- α positively correlated with the extent of type-I IFN signalling identified by scRNA-seq (data included in **Supplementary Fig. S10b-e**). The extent of type-I IFN signalling was determined by generating a “IFN score” on a per cell basis using the Seurat “AddModuleScore” function, with genes in the “type I interferon-mediated signalling pathway” GO term (GO:0060337). Per sample scores were generated by either summing the scores for the cells in a sample or taking a mean (**Supplementary Fig. S10 b, c**, respectively; the figure legend is included in lines 1118-1126; methods section: lines 969-974). We have added the type -I IFN signalling data (IFN scores) for the different cell clusters in **Fig 7c** and in the results section (lines 461-468).

Point 7. The authors did not specify catalog or clone numbers for the antibodies or reagents used in the study, particularly for PD-1/PD-L1 blocking antibodies.

This information can be found in the Key resources table included in the submission.

Point 8. The DENV2-specific T-cell function analysis is unclear. After NS3 DENV2 peptide stimulation, the authors reported that DENV2-specific T cells were predominantly monofunctional. It is unclear what is meant by "DENV2-specific T cells" here—does this refer to all stimulated T cells? Is there literature supporting that T cells after DENV2 peptide stimulation are truly DENV2-specific?

Intracellular cytokine staining (ICS) of PBMCs after stimulation with and without peptide pools comprising of peptides spanning viral protein sequences is a validated and widely used method for detection of virus-specific T-cells (e.g., Lamoreaux et al. Nature Protocols 2009). This method has also been widely used to detect dengue-specific T-cell by us and others (Duangchinda et al PNAS 2010; Rivino et al J Virol 2013; Rivino et al Sci Transl Med 2015).

When we refer to “DENV2-specific T-cells” we are referring to the T-cells that produce cytokines following stimulation with NS3 peptide pools from DENV2 (see Fig 4a, shown for CD4⁺ and CD8⁺ T-cells upon stimulation with or without DENV-2 NS3 peptides or with a PMA/ionomycin as a positive control). To avoid confusion, we have now called these cells “NS3 DENV2-specific T-cells”.

Are these cells related to Ki67+CLA+ bona fide DENV-specific T cells? We have previously shown that in dengue infection Ki67 and CLA co-expression can be used to distinguish DENV-specific T-cells from “bystander activated” T-cells, which are Ki67⁺ but lack CLA expression (Rivino et al Sci Transl Med 2015). Therefore, in the current paper we called these cells “*bona fide* DENV-specific T-cells”. To avoid confusion we have now eliminated the term “*bona fide* DENV-specific” T-cells and call these cells Ki67⁺ CLA⁺ T-cells (see lines 164-194; 221-222; 251; 274; 298).

Furthermore, how does the author define T-cell function using Boolean gating? Does the presence of IL-2 in these cells indicate that they only express IL-2, or could they be producing

other cytokines as well? Boolean gating is performed using flowJo to define the number of cytokines produced by NS3 DENV2 specific T-cells. We have included a figure showing our boolean gating strategy (**Supplementary Fig. S3a**). Data for each cytokine (including IL-2) is shown in **Fig. 4d**, e indicates the total percentage of CD4⁺ and CD8⁺ T-cells that are producing the indicated cytokine - these cells may be producing other cytokines as well.

Point 9. Did the authors consider or apply corrections for multiple testing in their statistical analyses?

Yes, all data was corrected for multiple testing using FDR correction or the Benjamini-Hochberg correction- please refer to each figure legend for the test used and to “Data and statistical analyses” in the Methods section (lines 998-1012).

Point 10: In the NK-cell killing assay, PBMCs were co-cultured with K562 cells. However, this experiment does not directly demonstrate the effect of K562 stimulation on NK cells. To strengthen the results, the authors should co-culture sorted NK cells with K562 cells and perform subsequent experiments.

Here we use the K562-based NK-cell killing assay which is the gold standard assay to assess NK-cell cytotoxic function in human PBMCs (e.g., Alter et al Journal of Immunological Methods 2004). In this assay CD107a expression in NK-cells correlates with NK-cell mediated lysis of target cells. This assay is performed using total PBMCs and does not require purification of NK cells which is not feasible in many studies including ours, utilising PBMCs from hospitalised and paediatric patients, due to the limited blood volume we can take from these patients. The same K562-based NK-cell cytotoxicity assay has been successfully used by us previously to assess NK-cell cytotoxicity in Singaporean dengue patients (Zimmer et Nat Comm 2019).

Point 11: The authors used classic marker-based tests to explore the association between phenotypic and functional impairment of CD4⁺ T, CD8⁺ T, and NK cells with SD in patients enrolled at the onset of SD. While partial validation experiments were performed, many of the descriptions of phenotypic outcomes appear overstated. These claims should be reconsidered or clarified.

Thank you for this comment. We have reconsidered and clarified statements associating phenotype with functional impairment and toned down our conclusions. We have changed “cytotoxic capacity” to “cytotoxic potential” when this is based on a phenotypic trait (granzyme B expression; for e.g. in the abstract, line 37). In lines 93-96 we have clarified the nature of the differences in the phenotype of T and NK-cells in SD versus non-SD dengue and state these features suggest immune dysfunction of these cells.

Point 12: The manuscript requires in-depth editing to address grammatical and formatting issues.

We thank the reviewer for highlighting this – we have now made in-depth edits to the manuscript and figure legends.

Reviewer #2 (Remarks to the Author):

Dear authors,

The manuscript “Early NK-cell and T-cell dysfunction marks progression to severe dengue in patients with obesity and healthy weight” was reviewed. The authors evaluated T and NK cells function from PBMC of dengue infected patients with different severity outcomes using mainly multi-color flow cytometry, ex-vivo stimulation with NS3 overlapping peptides with some additional plasma mediators, metabolic and scRNASeq. The strengths of the study is the high number of samples in the cohort and functional analysis including NS3 overlapping peptide stimulation. However, the sample collection timing differences between samples and confounding factors should be considered. The authors conclude that SD has dysfunctional NK and T cells potentially leading to severe disease but the data is not all conclusive. The presence of higher PD-1 T cells is evident but in functional study, NS-3 specific CD8 T cells in SD has higher CD107a suggesting they might be functioning. Therefore, interpretation of dysfunctional T cells based on PD-1 expression should be taken with caution. scRNASeq data is presented in the last figure but with limited result, the authors should be able to dive more in-depth in the analysis to provide more in-depth understanding of potential molecular mechanisms of the findings found by FACS and DV peptide stimulation.

Major comments

1. Sample collection timing and potential confounding factors. Sample collection timepoints T1 span several days, as immune response to DV is known to be highly dynamic during febrile phase and at critical phase. It is not clear from table S1 which day in each donor is considered as T1 and T2. It is also not clear whether T1 is during febrile phase in all donors or also include day after fever subsided. There is one sentence mentioned that SD include both day 3 (likely during febrile phase) and day 5 (in critical care, likely during critical phase). It is also not clear which clinical stage of T2 is. It was mentioned, T2 is approximately 3 days after T1, do patient still has fever at this point? Is it the same in all patients?

Thank you for this comment and we apologise for the confusion. In dengue the febrile phase coincides with the viraemic phase with both fever and viraemia waning at day 5 from illness onset (see schematic on the right showing the course of a secondary dengue infection; day of illness is calculated from fever onset). Hence timepoint 1 (TP1; ≤ 5 days) captures patients in the febrile phase while patients are in the post-febrile phase at TP2 (days 6-9). We have now included these details in Fig 1a, and in the results section (lines 109-111).

It might be clearer to analyze the result day by day in relation to the day of fever subsided comparing between disease severity or at least consider day of sampling as potential confounder and perform statistical analysis accordingly. This is of concern because SD T1 sample collection seems to be later (majority of samples were collected on day 5 vs Non SD, majority were collected

on day 2-3 after symptom onset) (Sup table 1). This might affect the results particularly IFN I is known to upregulate early and go down afterward.

Thank you for highlighting these valid points which we have carefully considered when analysing our data. Severe dengue manifestations occur around days 4-5 from illness onset in approx. 5% of patients; this has limited the number of severe dengue cases included in previous studies recruiting patients at the time of illness onset. Here, to include a higher number of severe cases (30 out of 150 total patients= 20%) and allow more robust comparisons of immune responses across disease severities, we recruited patients up to day 5 of fever onset. This allowed us to identify patients that were already presenting with symptoms of severe dengue as well as those that progressed to SD during the course of the study. In the linear discriminant analyses shown in

Fig. 1j-m, the average day of illness of patients at TP1 is 4.3 for SD patients and 2.9 for non-SD patients. To ensure our data was not skewed by datapoints from SD patient samples taken at days 4-5 of illness onset we performed the same analyses including only data from days 3-4 (TP1) for both severe and non-severe dengue groups. This analysis (shown on the left)

shows a clear separation between the patient groups, although fewer SD patients were included – these comprised of one patient who had already progressed to SD at the time of admission (SD: filled red circle) and three patients who progressed to SD during the course of the study (SDp= empty circles).

We also reanalysed the data from all patients at both timepoints (now included in **Supplementary Fig. S2a**; and copied below), highlighting patients in the severe group who had already progressed to SD at admission (SD patients: 1 patient at day 4 and 6 patients at day 5; filled red circles) and those who progressed to SD during our study (SDp= SD progressors: 2 patients at day 4, 1 patient at day 3 and 1 patient at day 2; empty circles). T and NK-cell profiles of SD and SDp patients are overlapping and clearly segregate from those of patients with non-severe dengue. Analyses of immune profiles by day of fever was not informative due to the small sample set available per day of fever.

Supplementary Fig. S2a.

As OW/OB seems to have more severe disease, how could we tease out the effect of body weight vs severity on immune profiles? Other additional potential confounders should be included in table S1 eg. DV serotype and viremia level at T1 and T2, body weight in different severity groups. Proper statistical test should be performed to evaluate any confounding effects of these variables.

Overweight/obese patients were matched by age, sex and illness phase to those with healthy weight and within the severe group we had 16 patients with healthy weight and 14 patients with overweight/obesity (see **Supplementary Table S1**)- hence there was an approx. equal representation of BMI groups within the severe dengue group.

The DENV serotype of infection is included in **Supplementary Table S1**. There is a similar distribution of DENV serotypes between SD and non-SD with the majority of infections being DENV2, following by DENV1, in line with the predominant co-circulation of these two DENV serotypes in Vietnam. There were no DENV3 cases and a minority of DENV-4 cases (approx. 15-20% of patients). Analyses of data including only DENV2 cases shows similar results to what is presented in our manuscript.

Viraemia was measured at admission (TP1); however, we do not have information as to whether these measurements were taken at the time of peak viraemia for each patient. Therefore, analyses of viremia between patient groups is not informative for this study. Viraemia is not detected at TP2 as it typically clears around day 5 of fever onset.

2. Fig 2h. If the author could annotate clusters (using combined data) before describing details, it would be easier to understand the biological significance of the findings

Thank you for this comment. We agree that including all the annotated clusters makes it easier for the reader to understand the data. We have now changed **Fig. 2h** and **Fig. 3k** to include all clusters (also shown here on the right). We believe showing the plot with combined data from SD and non-SD patients does not provide additional information (indicated here as “All samples”), hence we have not included the latter plot in manuscript figure.

3. Line 205 “HLADR+CD38+CD8+ T correlate with plasma levels of leptin suggesting that obesity could have impact on CD8 T cell activation”... but Fig 3 F , more directly, show that OW/OB not correlate with CD8 activation. Please be careful not to over interpret the result. Thank you for this observation, we have deleted this statement.

4. NS3- specific T cells response

a. It would be important to know previously infecting serotype and analyse the data separating previous vs current infecting serotype. Test such as PRNT could be performed to identify previously infecting serotype.

It is challenging to accurately determine the serotype of the primary infection in patients experiencing secondary dengue due to the high cross-reactivity of antibodies elicited by 2 DENV serotypes towards 4 DENV serotypes. Hence, while PRNT analysis allows to accurately determine the serotype of infection in primary dengue, in our experience it does not allow an accurate understanding of the DENV serotype of infection in secondary dengue patients. In the HCMC area (Vietnam), there has been a predominant circulation of DENV2 and DENV1 over the last decade hence it is likely that DENV1 and 2 were the serotypes of the first infection in respectively the DENV2 and DENV1-infected patients, but this can only be inferred (Phoung et al. PLOS Negl Trop Dis).

b. As antigen-specific T cell response depends on HLA, have the authors adjusted of donors' HLA?

For our analyses we used pools of 15-mer overlapping peptides which are not restricted to any specific HLA type therefore allowing to capture “universal” T-cell responses restricted to all HLA types. A comparison of T-cell responses across disease severities per HLA type would require a larger dengue cohort which includes an approx. equal number of patients expressing each HLA type of interest. While this is a very interesting point which we are addressing in a new project we are now starting, it goes beyond the scope of this study.

c. Fig 4c high variability is observed.

Thank you for this comment. Our data is in line with previous work showing that T-cell responses are highly variable across dengue patients (e.g., Duangchinda et al PNAS 2010; Rivino et al. J Virol 2013; Chng et al Immunity 2019). While highly variable, our data show that these responses are overall significantly higher in patients with non-SD compared to SD at this timepoint.

d. It would be interesting to further investigate which epitope of NS3 are dominant.

Thank you for this comment. Previous studies by us and others have investigated the DENV epitopes across the all DENV proteins including NS3 (e.g., Rivino et al J Virol 2013; Weiskopf et al.PNAS 2013). While it would be interesting to perform similar analyses in our patient groups, the limited amount of blood we could obtain from these patients made it necessary for us to focus on the main and novel question of this study.

e. It would be of great interest if scRNASeq could be performed after NS3 stimulation.

We agree with the reviewer that this would be interesting however due to limiting cell numbers we chose to focus our scRNA-seq analysis on the investigation of PBMCs *ex vivo*, as this does not involve manipulation of cells which could skew gene expression. Of note, these PBMCs are obtained from patients with dengue and therefore have already been stimulated *in vivo* with DENV antigens.

f. Line 253 “SD patients also display increased frequencies of degranulating CD107a+” . This could suggest that NS-3 specific CD8 T cells in SD, even though they express high PD-1 and inhibitory co-receptors, they are functional. Consistent with previous report in COVID that PD-1-expressing SARS-CoV2-specific CD8+ T cells are not exhausted but functional. (<https://www.sciencedirect.com/science/article/pii/S1074761320305094>)

Thank you for this comment. Our data suggests a distinct kinetic of expansion of NS3 DENV-specific T-cells in SD compared to non-SD patients, whereby T-cells are strongly activated in SD patients at TP1 (**Fig 4b**) possibly due to the higher viral loads known to occur in SD. This could explain the higher CD107a levels at TP1 in SD, which is however not observed at TP2 where there is an opposite trend of lower NS3 DENV-specific T-cells in SD patients. Our data show CD4⁺ and CD8⁺ T-cell expression of a range of co-inhibitory receptors beyond PD-1 (**Fig. 5**), a phenotype which is consistent with decreased T-cell functionality and T-cell exhaustion. Further studies are needed to address in detail the molecular mechanisms occurring in these cells.

5. Fig 5k, data is highly variable, only 2 in 9 donors showed increase CD8+GzB+Perf+ upon PD-1 blockade, please perform statistical test including all samples. It is difficult to draw conclusion from the data presented.

Thank you for this comment. We agree and have acknowledged that PD1/PDL-1 blockade is effective *in vitro* for only some dengue patients, with variation across patients (line 377). The statistical test we performed considers all samples (one-way ANOVA with Benjamini-Hochberg correction) and shows that overall there is a statistically significant increase in granzyme B⁺

perforin⁺ CD8⁺ T-cells after blockade. We believe it is important to show this data, acknowledging the variability across patients. Of note, PD1/PDL1 blockade has represented an important advancement for cancer treatment despite the mixed results showing efficacy in only a proportion of patients.

6. Figure 7. scRNASeq

- The limitation of probe-based BD rhapsody without TCR/ BCR data should be discussed
- A lot more in-depth analysis of scRNASeq data can be done and would potentially provide more in-depth understanding of the molecular mechanism of T and NK cell response as well as cell-cell communication with other immune cells within PBMC. DEG in other cell clusters should also be described. The authors should not only limit their analysis to differential cluster abundance and GO term in all clusters combined.

- Limitation of the approach: We thank the reviewer for this comment and have included these considerations in the results and discussion sections (lines 469-471 and 539-540). We chose the probe-based BD Rhapsody approach for complex biosafety reasons as DENV is a hazard group 3 pathogen in the UK and we are currently not able to perform 10X scRNA-seq in our containment level (CL)3 laboratory. We recognise the limitations of the probe-based BD Rhapsody approach which limits our analyses to ~500 genes and does not provide a comprehensive view of the gene expression landscape. We are also unable to identify specific TCRs/BCRs or examine repertoire diversity. However, this analysis provides robust supportive data in a unique patient sample set and allows us to answer a specific question around type-I IFN responses in patients with dengue.

- Further analyses of the data: We performed differential expression analysis (Wilcoxon rank sum test) between severity groups for each cell type individually followed by pathway analyses. For some cell types, there were very few differentially expressed genes (e.g., MAIT cells and pDCs). For CD4⁺ T-cells, CD8⁺ T-cells and NK-cells, the list of DEG is strongly enriched for genes involved in type I IFN signalling and pathway analyses (over representation analyses or ORA) of these cells shows similar results as for total PBMCs. We have added the ORA plots for CD4⁺ T-cells to those of CD8⁺ T-cells and NK-cell in **Supplementary Fig S9a-c**.

As explained to reviewer 1 (point 6), we also quantified the extent of type-I IFN signalling across cell types by generating a type-I IFN score based on expression of genes within the GO term “type-I interferon-mediated signaling pathway” (**Fig.7c**). These data shows that Type-I IFN signalling scores were decreased in CD4⁺ T naïve/central memory (TCM) and CD4⁺/CD8⁺ T effector memory (TEM) cells, naïve B-cells NK-cell, MAIT cells, proliferating lymphocytes and monocytes from patients with SD compared to non-SD.

We performed CellPhoneDB cell-cell interaction analysis to gain insights into cell communication. CXCR3-CXCL10 is the main molecular interaction that is differentially regulated in SD compared non-SD (downregulated in NK cell and CD4⁺/CD8⁺ T-cell subsets, data shown below). As CXCL10 is an IFN-induced gene, these results are in line with decreased type-I IFN signalling in SD

CXCL10-CXCR3
DOWN IN SD

CXCL10-CXCR3
UP IN SD

7. The authors claim that IFN-I is impaired in SD, based on scRNASeq data. As in comment 1, it is important to align timepoint of sample collection between groups for fair comparison. Further, protein level confirmation such as serum level of IFN-I and intracellular cytokine staining of IFN-I would be crucial.

We have now validated the type-I IFN signatures at the protein level using TMT mass-spectrometry based proteomics and Luminex - see point 6, Reviewer 1 for a detailed response;

manuscript lines 469-488 and **Fig. 7d-e**; Fig.7 legend (lines 808-816). Briefly, our proteomics data confirms the downregulation of proteins involved in type-I IFN signalling. We also show using Luminex that plasma IFN- α levels are significantly decreased in severe compared to non-severe dengue patients at TP1, and these levels correlate with the extent of type-I IFN signalling detected by scRNA-seq (**Supplementary Fig. S10**).

Regarding the day of illness, please refer to our response to your point 1 and reanalyses of data in **Supplementary Fig S2a**). For the new proteomic analyses and the scRNA-seq analyses the average days of illness onset are respectively: 4.4 for SD and 3 for non-SD and 4.3 for SD and 2.9 for non-SD. While we acknowledge the limitation of these analyses that do not compare type-I IFN levels at the same day of illness (see lines 542-548), our data supports the following published findings. A study in a Cambodian cohort (Upasani et al. *Frontiers Immunol* 2020) shows decreased serum IFN- α and IFN- β levels in patients with dengue haemorrhagic fever (DHF)/dengue shock syndrome (DSS) compared to dengue fever- the mean day of illness for these patients was respectively 3.5 and 3.4. The study by Pichyangkul et al. *J. Immunol* 2003 shows a decrease in IFN- α producing plasmacytoid DCs in patients that progressed to SD compared to non-SD patients (day of illness <72 h).

Minor comments

1. Please show FACS gating strategy with FMO for main Fig 2 and 3 in supplementary data.

Please find our gating strategy included in Supplementary Figure S1. As we had limiting PBMC samples available for these patients and the markers we analysed had a clear negative and positive population, we did not include FMOs here.

2. Ki67+CLA CD4+ T cells probably not enough to be called “Bona fide DENV-specific T cells” as antigen specificity is not directly shown. (line 148)

As mentioned above we have now named these cells Ki67⁺CLA⁺ cells rather than “bona fide” DENV-specific T-cells.

3. All data should be deposited and allow access to audience.

All flow cytometry data has been deposited at www.flowrepository.org. IDs are included below and indicated in the data availability section (lines 1035-1036).

ID: FR-FCM-Z8DN

ID: FR-FCM-Z8E3

ID: FR-FCM-Z8EU

ID: FR-FCM-Z8EV

ID: FR-FCM-Z8EG

Reviewer #3 (Remarks to the Author):

This paper by Gregorova et al describes early NK-cell and T cell dysfunction in patients which develop to severe dengue. The paper is build around an in depth phenotypic analysis of dengue patients, and compares phenotypes at different timepoints in the disease, between non-severe and severe dengue and between overweight/obese patients and patients with healthy weight. With all these comparisons, the paper feels diluted in its messages and lacks clear

conclusions. It also makes the figures sometimes difficult to understand.

We thank the reviewer for acknowledging the depth of analyses performed. We have made extensive edits to the text and hope that it is now clearer that the main message we want to convey is the presence of dysfunctional T and NK-cell responses and a blunted type-I IFN response in severe compared to non-severe dengue patients. This is a unique dengue cohort which also allows us to evaluate the impact of BMI on the immune response to dengue hence we believe it is important to include comparisons of immune responses between patients with healthy and high BMI, although we realize it adds complexity to the study.

Major concerns:

1. Dengue disease is very dynamic, and changes in immune cell composition occur rapidly. Patients are at risk to develop severe disease in a small window after onset of symptoms, usually 3-6 days post onset of symptoms (WHO guidelines, 2009). Therefore, including patients up to 5 days of fever in the SD group and not in the non-SD group will introduce a large variability in the immune cell phenotypes and functions. Looking at Table S1, only 13 non-SD were included at day 4 and none later (14% of the non-SD patients), while in the SD group 21 patients were enrolled at day 4 or later (72% of the SD patients).

Thank you for this comment. Please refer to our response to reviewer 2 point 1 for a detailed response. While we agree and acknowledge this is a limitation of the study (lines 542-548), our new analyses show that our results are robust and not biased by the differences in day of enrolment. In summary we show: (i) T and NK-cell profiles remain distinct in SD versus non-SD patients when selecting only data from days 3-4 of illness onset (figure provided to reviewer 2 point 1); (ii) similar T and NK-cell profiles in SD patients who were recruited prior to progression to SD or had already progressed to SD at admission (**Supplementary Fig. S2a**).

2. A major concern is the interpretation of the data and the conclusion that the authors have identified prognostic signatures before the onset of severe dengue disease. However, the critical phase is between 3-6 days of illness.

We thank the reviewer for this comment. Studies show that the critical phase in dengue occurs around time of defervescence/late febrile phase (days 4/5-7 from fever onset, <https://www.cdc.gov/dengue/hcp/clinical-signs/index.html>). Our analyses for TP1 include samples at day 3-5 so largely before/at the onset of SD while TP2 includes samples at days 6-9. Linear discriminant analyses (LDA) in **Fig. 1 j-k** includes eleven SD patients: four of these were recruited prior to development of SD (two at day 4, one at day 3 and one at day 2; defined as SD) and seven were classified as SD at admission (one patient at day 4 and six patients at day 5; defined as SD progressors=SDp). In **Supplementary Fig. S2a** we show that T/NK-cell profiles are overlapping in SD and SDp, therefore in the SD patient group the T/NK-cell features we describe are already present prior to the onset of SD. For major clarity we have added a schematic explaining the disease course (viraemia and severe dengue phase) to **Fig. 1a**.

Can the authors confirm their most important findings when selecting only patients before the critical phase? This analysis is also needed if the authors want to conclude on NK and T cell dysfunction in patients progressing to severe dengue, as stated in the title, abstract and discussion. In case that is not possible due to the low number of patients in the severe disease

group, the title, and interpretation should be rephrased, refraining from concluding these signatures are prognostic markers for SD.

Thank you for this comment. Please refer to **Supplementary Fig 2a** and data shown in responses to reviewer 2, point 1.

3. Similarly, previous immune history has a major impact on immune cell phenotypes and function. Line 702 mentions 5 cases of primary infection in the non-SD group, could the authors confirm the most important findings when excluding these primary infections?

We have reanalysed the data and confirm that our major findings remain valid when excluding the primary dengue cases. We include below the reanalysed data for the following figs:

- **Fig 1j-k and l-m** (LDA of T and NK-cell profiles)
- **Fig. 2 a-b** (activation and PD-1 expression in CD4⁺ T-cells)
- **Fig 3 a, b, d and e** (activation, PD1 and granzyme B expression in CD8⁺ T-cells)

The following analyses only included patients with secondary dengue infection: **Fig 5 a, b**: expression of co-inhibitory receptors by CD4⁺ and CD8⁺ T-cells); **Fig 6**: NK cell analyses and **Fig. 7**: scRNA-seq and proteomics analyses.

Fig 2 a,b reanalysed to include only secondary cases

Fig. 3 a, b, d, e reanalysed to include only secondary cases

4. the gene expression data are obtained from a gene expression analysis of a targeted set of genes. This experiment has other important limitations, such as the pooling of 12 individuals per cassette. Both of these important informations should be mentioned in the results section before describing these results.

As mentioned in reply to reviewer 1 point 3 and reviewer 2 point 6 we have acknowledged the limitations of the BD Rhapsody approach (lines 469-471 and 539-540) which was chosen due to complicated biosafety reasons as DENV is a hazard group 3 pathogen in the UK so all experiments with dengue patient samples need to be performed in a containment level 3 laboratory. As this approach uses a targeted panel (~500 genes), the data does not provide a comprehensive view of the gene expression landscape, but it remains valuable to address specific hypothesis-driven questions such as whether disease severity associates with changes in type-I IFN signalling. Moreover, we have now confirmed our scRNA-seq results using a holistic proteomics analysis- these new data confirm the downregulation of type-I INF responses in SD patient PBMCs (Fig. 7 d-f and lines 471-482)

In our scRNA-seq approach, cells from each patient were labelled with a different Sample Tag and we identified the patient source of the cells during analysis through demultiplexing, hence pooling 12 individuals per cassette does not represent a limitation in our analyses.

5. Samples were pooled on only 2 BD Rhapsody Cartridges, 3500 cells each, which is over the maximum capacity stated by the company (around 40000, see e.g. <https://www.bdbiosciences.com/content/dam/bdb/marketing-documents/products-pdf-folder/instruments/bd-rhapsody-express-single-cell-analysis-system/BD-Rhapsody-Single-Cell-Analysis-System-Brochure.pdf>). Thus it is crucial to add tables and QC plots to show basic statistics of the scRNA-seq analysis, such as the number of cells before and after filtering steps, the number of identified doublets, the distributions of the number of RNA molecules per cell, the number of different genes per cell, the rate of mitochondrial genes and the cell cycle stage of the cells for quality control purposes.

The maximum cell capture for the BD Rhapsody Cartridge is 40,000 cells, which requires loading of > 40,000 cells – see page 4 of <https://www.bdbiosciences.com/content/dam/bdb/marketing-documents/products-pdf-folder/instruments/bd-rhapsody-express-single-cell-analysis-system/BD-Rhapsody-Single-Cell-Analysis-System-Brochure.pdf>. We have followed the manufacturer’s instructions. As explained above we used a probe-based BD Rhapsody approach which includes a set number of genes – therefore we were not able to investigate expression of mitochondrial genes.

6. The differential gene expression analysis was performed using the non-parametric Wilcoxon Rank Sum test, which is the default method used in Seurat (originally for performance reasons). While Wilcoxon has been indicated by a study by Jingyi Jessica Li group (Li et al., 2022: 10.1186/s13059-022-02648-4) as the best method for controlling FDR, the results of this study remain controversial in the community (see e.g. https://www.reddit.com/r/bioinformatics/comments/tg8v7z/paper_exaggerated_false_positives_by_popular/ and https://www.reddit.com/r/bioinformatics/comments/161p3xx/wilcoxon_rank_sum_test_as_alternative_to_deseq2/). In addition, several other comparative studies (e.g. Miao and Zhang 2016: 10.1007/s40484-016-0089-7, Sonesson and Robinson, 2018: 10.1038/nmeth.4612, Li et al., 2022: 10.1371/journal.pone.0264246) rather indicate the use of methods such as DeSeq2, which considers sample technical heterogeneity, as preferred. Therefore, could the authors reanalyse the data with a method which considers sample technical heterogeneity, such as DeSeq2, as e.g. shown in the Seurat vignette (https://satijalab.org/seurat/articles/de_vignette).

Thank you for this comment. We performed pseudobulk differential expression analysis using edgeR. Fewer genes were identified as differentially expressed compared with the Wilcoxon rank sum test (for e.g. for all cell types combined, 62 genes were differentially expressed between severity groups with edgeR and 115 with the Wilcoxon rank sum test). Pathway analysis shows downregulation of the GO term “type I interferon-mediated signalling pathway” in severe dengue for both edgeR and Wilcoxon confirming that results are consistent between the two differential expression analysis approaches (data shown below). Moreover, the proteomics analyses led to similar results highlighting a strong representation of genes involved in innate response and type-I IFN signalling amongst the downregulated genes in SD.

Minor

concerns:

Figure 1a should include immunophenotyping below single-cell RNA sequencing, as most data (Fig1-6) are actually generated by flow cytometry.

Figure 1 includes only flow cytometry data. ScRNA-seq data is included in Fig 7. We have now clarified this in the Fig. 1 legend (lines 642-643).

Figure 2i and 3l, this reviewer does not understand what is shown in the top part of the figure (%events on y axis) or what the arrows indicate? maybe a more clear description or figure can help.

We apologise for the confusion and have now clarified this in the Fig. 2 and 3 legends (lines 676-679; 701-704). For example, for Fig 2i: “Stacked bar chart showing the frequency (y-axis) of each cluster in the patient groups with the bubble graph representing MFI levels (colour scale) and cell frequencies (size). The clusters are indicated on the x-axis; arrows highlight selected clusters on the top of the graph”.

Line 430-430: comparisons throughout the paper are made between SD and non-SD, could the authors expand on the results supporting this statement?

The most recent WHO guidelines classify dengue into dengue, dengue with warning signs and life-threatening severe dengue (WHO 2009, Dengue: Guidelines for Diagnosis, Treatment, Prevention and Control). Data for these groups is shown separately in Fig j-k but for most analyses we merge data for “dengue” and “dengue with warning signs” into one group named “non-severe dengue” as we did not detect striking immunological differences between these two groups. Defining immunological biomarkers that allow the early identification of patients before they progress to severe dengue would greatly improve patient management as it would allow clinicians to prioritise monitoring of patients at high risk of progressing to severe dengue while allowing to discharge patients not at risk. The high burden of dengue cases during the dengue

season in dengue-endemic regions remains a challenge as hospital beds and resources are limiting in these regions.

Line 677 mentions “a proportion of severe patients were admitted up to day 5 of fever”. This is also shown in Figure 1a. It should be added how many patients this is, both in the methods text and the figure (data can be found in table S1)

We have included details in the methods section, lines 832-838 for all patient samples. For Fig 1, data is broken down into SD and SD progressors in **Supplementary Fig S2a**, with the number of patients for each group indicated in the legend (lines 1060-1064).

Reviewer #4 (Remarks to the Author):

NCOMMS-24-65884 (Gregorova et al)

In this study Gregorova et al have comprehensively characterized the phenotypic, functional and transcriptional analyses of T and NK-cells in a cohort DENV-infected subjects during the course of acute infection. This study offers a unique insight into T/NK cell biology during DENV-infection in the context of weight/obesity stratification. The study is well-designed and well-presented and an important contribution to the field. Pending the authors addressing several minor issues satisfactorily, I would recommend this paper for publication:

We thank the reviewer for the supportive comments.

1. In general, it is unclear how the breakdown of HW versus OW/OB stratifies between those under 18 and those over 18 (i.e. children versus adults). I'm assuming the OW/OB group falls more within the adult (over 18) age-group. Hence, while still very interesting, is the phenomenon presented more of a children-versus-adult finding or age-related? Can age be taken account of using a multi-variable confounder analysis?

Healthy weight (HW) patients were age and sex matched to the overweight/obese (OW/OB) patients, hence the median age of patients in these two groups is the same and age is not a confounder factor for the analyses of immune responses between these BMI groups (median age for both groups is 15 years, see **Supplementary Table S1**). Dengue patients in the HW and the OW/OB groups include both adults and children (defined in this study as \$\geq 19\$ and \$< 19\$ years, respectively). The breakdown of the OW/OB and HW groups is as follows:

- 60 patients with HW: children n=40; adults n=20. Median age: 15 years (interval 10-30).
- 64 patients with overweight/obesity: children n=42; adults n=22. Median age: 15 years (interval 10-30).

We performed further analyses to evaluate whether age is impacting the observed NK/T-cell signatures in this cohort. Linear discriminant analyses of data from Fig. 1 shows: (i) similar clustering of immune signatures according to disease severity if we take data only from paediatric patients (**Supplementary Fig. 2b**); (ii) tight clustering of data from paediatric and adult patients demonstrating that we are not seeing an age-related effect (**Supplementary Fig. 2c**). These considerations have been added to the results section (lines 145-147).

2. Figure 1B: What is the denominator for the presented frequencies (percentages)? Total lymphocytes? All PBMC? Since percentages are essentially a zero-sum game, increases in one or more phenotypes are associated with similar decreases in others.

We apologise if this is unclear, we have now added our gating strategy for the different populations in **Supplementary Figure 1a, b**. The frequency of CD3⁺, CD3⁻ and CD56⁺CD16⁻ cells (total NK-cells) is calculated from live cells and from the singlets gates; the frequencies of NK-cell populations are calculated from the CD3⁻ gate; the frequencies of CD4⁺ and CD8⁺ T-cells are calculated from the CD3⁺ gate.

3. Lines 133-134. It is stated that “T/NK-cell profiles are more strongly impacted by clinical outcomes than BMI”. This would suggest that T/NK cell profiles are reactive to the infection outcome than driving it perhaps. Is this what the authors are really trying to say here? This is important as it gets at the question of cause and effect, which is the core question in such studies. It certainly appears from subsequent data that NK cells are more likely the drivers (if at all) rather T cells.

We thank the reviewer for pointing this out and we agree the wording used is misleading. We have deleted this statement.

4. Lines 147-148: Using Ki-67 and CLA co-expression as a surrogate for DENV-specific T cells is fine but using the term bona fide DENV-specific T cells is a stretch without having an example of confirmation of the specificity.

We thank the reviewer for this comment. As explained in response to reviewer 1 point 8, we have previously shown that Ki67 and CLA co-expression can be used to distinguish DENV-specific T-cells from “bystander activated” T-cells, which are Ki67⁺ but lack CLA expression (Rivino et al Sci Transl Med 2015). However, since we are not demonstrating this in the current paper and to avoid confusion we have now eliminated the term “*bona fide* DENV-specific” T-cells and call these cells Ki67⁺ CLA⁺ T-cells (see lines 164-194; 221-222; 251; 274; 298).

5. Lines 161-162: The terms protective role and detrimental role are only suggestive. Those phenotypes could be resultant of the outcome, and not necessarily driving it.

We agree that we are showing a correlation here rather than demonstrating causality. We have reworded this to “These data show a correlation of PD-1⁺ CD4⁺ T-cells with SD and of a cytotoxic CD4⁺ T-cell response with uncomplicated dengue” (lines 191-194). We also added a statement to highlight the cause and effect aspect in the discussion (lines 560-562): “However, the cause-effect relationship between NK/T-cell dysfunction and dengue clinical outcomes remains to be determined”.

6. Figure 2 panel H is a little confusing: It looks like the total number of subjects T1 = 42 while the total at T2 is 84. Were any of these the same subjects? I’m assuming that they were. It is understandable with such studies that getting matched samples for subjects is difficult. For the SD subjects at T1 the sample numbers are lower (n=11) than the others. Is this why the dot-plot appears less dense? Could it be that some clusters have decreased, rather the identified clusters having simply increased?

We thank the reviewer for this comment. For more clarity we have now visualised all the 13 clusters in **Fig 2h**. All clusters that display changes in frequencies between SD and non-SD patients are displayed in **Fig 2j**. Below we show graphs for all 13 clusters to highlight that the decrease in frequencies of CD4⁺ T-cells in some clusters are not a consequence of the increase of frequencies in other clusters.

7. The directly comparative analyses of CD4⁺ and CD8⁺ T cells shown in Figures 2 and 3 is greatly appreciated and clearly highlights the important features and differences between the two lineages and between T1 and T2 sampling times.

We thank the reviewer for this supportive comment.

8. Could the functional data in Figure 4F be presented more like the phenotyping data in Figure 5E? It looks like the Spice application (or something similar) was used to generate the figure. The functional data may be more understandable in this format and is buried in Figure S3.

Thank you for this suggestion, we have tried different visualisation methods, but the one included in **Fig. 4f** is the most understandable in our opinion and best communicates the message we want to give of the ratio between NS3-specific T-cells that are degranulating, producing IFN- γ /TNF- α or doing both, in patients with SD versus non-SD.

I do understand that figure space may be limited though. Was any kind of sub-analysis of the functional studies performed to affirm using Ki-67 and CLA co-expression as a surrogate for DENV-specific T cells?

We are sorry for the confusion here, as also mentioned above in response to your point 4 we have now eliminated the term *bona fide* DENV-specific T-cells.

9. Lines 257-259, and a more general critique: Since the T cell functional response increases

disproportionately in SD compared to non-SD patients is it not simply possible that early and more rapid viral replication and hence antigen load is driving this T cell response? The SD outcome phenotype may be “set” earlier and the T cell response is reactive to this. Better innate control leads to development of an appropriately developed T cell response phenotype while an “overwhelmed” innate response results in driving a dysfunctional and exhausted T cell response.

Thank you for these insightful analyses- we agree that the higher viral load present in SD patients would lead to increased antigen presentation to T-cells and possibly excessive T-cell activation while in non-SD a more adequate innate response (e.g., type-I IFNs and NK-cells) will lead to a more efficient control of virus replication and a more balanced and effective T-cell response. We have reworded this section in the discussion to better convey this message (lines 563-567)

10. Lines 321-323: Since the PD-1 data supports glycolysis as the preferred metabolic usage pathway in OB/OW versus HW cells, does the corollary hold true: i.e. in HW patients T cells are more prone to using anabolic metabolism? The methodology used here allows for a crude analysis of metabolic pathway usage and perhaps a more in-depth metabolic pathway analysis is required to answer this. In general, the observation here is in line with PD-1+ cells being metabolically more quiescent and non-proliferative which makes sense.

Thank you for this comment; we agree that a more in-depth analyses of the metabolic state of these cells is needed to answer this question and gain more insights into metabolic pathways used by these cells.

11. Figure 6: The NK cell data is well presented, profound and of note. The main question I would have, is whether the observed NK cell deficit of function could be specific to the response to DENV-infection, or whether it is a generalized effect. i.e. if one were to analyze influenza or other viral infections (ssRNA viruses in particular) in the same way, would the same observation of NK cell impairment be true?

This is an interesting point- in general there is more data available for human NK-cell responses in chronic infections (e.g., HCMV and HBV) and less data for acute virus infections. Functionally impaired NK-cells were reported in severe COVID-19 patients, although these patients display elevated type-I IFN signalling which drives excessive activation of NK-cells, suggesting different mechanisms are involved in severe COVID-19 compared to severe dengue (e.g., Kramer et al. Immunity 2021). There are reports of NK-cell dysfunction occurring in influenza infection with several studies showing that influenza virus can infect human NK-cells and induce phenotypic changes and apoptosis of these cells (Mao et al. mSphere 2009; Guo et al. Immunol Cell Biol). To our knowledge no study to date has suggested that DENV can infect NK-cells.

12. It appears that overall innate sensing dictates the potential pathway to SD versus non-SD and that and genetic and behavioral influences such as in this case BMI and metabolism play a role in steering the outcome positively or negatively. One question which remains is relationship of the outcome T cell response and its effect on clinical outcome. This is still an outstanding question. It does not impact the quality of the study, but the authors should be clear that this is the case.

Thank you for these considerations, we have clarified this in the discussion (lines 560-562).

13. Lines 500-502: It is difficult to envision a scenario in which checkpoint inhibition could be applied in a systematic and effective way to modify DENV outcomes as a treatment strategy. How would one identify patients to treat when it seems like the effect is already set by the time patients would come to a clinic? Symptomatic treatment of plasma leakage still appears the best strategy.

Thank you for this consideration. Further studies are needed to confirm in larger patient cohorts that NK-cell features of dysfunction are detectable early in dengue infection - this would offer an opportunity to evaluate the use of these features as biomarkers to stratify patients at risk of developing SD. Availability of a biomarker for SD would allow close monitoring of patients and delivery of therapeutics early, when/if these become available. It remains unclear whether NK-cell dysfunction in SD is reversible and whether restoring NK-cell functions would lead to amelioration of disease.

14. While the scRNA-seq analysis is well performed it is unclear exactly what it adds to the study. For panel C, Figure 7, does this refer to all cells or is it “gated” or derived from a certain subset of cells within the PBMC? Perhaps Figure S7 shows this better as it focuses on CD8+ and NK cells, which were defined as important by the flow cytometry data. Since this scRNA-seq method (BD Rhapsody) is heavily dependent/biased on the pre-selected target genes it is perhaps not as powerful as drop-seq (such as 10X genomics) based technologies, which are agnostic and unsupervised.

We agree that the BD Rhapsody method is not as powerful as a whole-genome sequencing approach but as explained in response to reviewer 2 point 1 we were limited by complex biosafety issues as DENV is a hazard group 3 pathogen in the UK. However, we believe that the scRNA-seq analyses provides important information in a unique dengue patient cohort and reveals a downregulation of type-I interferon signalling in severe dengue, which could not be identified through flow cytometry. We now confirm the strong downregulation of type-I interferon signalling in SD using a proteomics approach which is a whole-proteome and unsupervised approach, therefore complementing the scRNA-seq data.

Fig. 7b (previously 7c) shows the results of over-representation analysis of genes downregulated in SD compared to non-SD for all PBMC cell types combined. As the downregulated pathways shown in Fig 7c are similar to those identified in NK-cells and CD4⁺/CD8⁺ T-cells (**Supplementary Fig. S9**), in the main figure we have chosen to show data for all cell types combined as it is representative of what is occurring across different cell types.

POINT BY POINT RESPONSE TO THE REVIEWERS' COMMENTS

Reviewer #1 (Remarks to the Author):

I appreciate the authors' efforts in addressing most of the comments and providing additional valuable data. However, I still have concerns about the response to Point 10 regarding the use of PBMCs rather than purified NK cells. PBMCs contain various immune cell types, which may confound the interpretation of NK cell-specific effects. The K562 cell line might also influence other immune cells, indirectly affecting NK cell function. Similarly, other related NK-cell killing assays should also consider this issue. Additionally, the authors cite a 2004 study, which may be outdated given recent advancements in NK cell research. I recommend considering more recent literature and adopting purified NK cell assays to provide more direct evidence of NK cell function.

We thank the reviewer for expressing their concerns. We agree and acknowledge in the discussion of our manuscript that lack of such experiments represents a limitation of our study (lines 480-483): "A further limitation of our study is the use of whole PBMCs and K562 target cells to assess NK-cell cytotoxicity, using a previously described approach³⁷. Further studies using purified NK-cells are needed to confirm the reduced cytotoxic capacity of NK-cells in SD compared to non-SD patients".

We cite our more recent publication (reference 37, Zimmer et al Nat comms 2019) that used the same approach (whole PBMCs) for functional analyses of NK cells in dengue patients.

Reviewer #2 (Remarks to the Author):

The authors addressed most of my concerns and performed additional experiments when necessary. Below, please find the list of minor comments by item listed in the initial revision.

1. Please explicitly state the limitation of different days (TP1) in SD and non-SD. Please also include the rebuttal analysis only day 3-4 (TP1) to supplementary.

We have explicitly stated in the manuscript the limitation of the different days for SD and non-SD patients for the flow cytometry (lines 477-480) and the scRNAseq/proteomics analyses (lines 510-514). We have added the data for day 3-4 only to Supplementary Fig. 2a.

Please add the viral load data of each donor on TP1 in supplementary table even though TP1 might not be peak viral load, but it is the time point in which PBMC were obtained for immunological analysis.

We have now added the viraemia levels in Table S1, showing these separately for patients with SD and non-SD dengue stratified by day of admission, as well as for total patients.

4a. Please add the explanation and limitation of identifying previously infecting serotype in the manuscript

We have added this explanation in lines 551-554: “Here we investigated DENV-specific T-cell responses in samples from secondary dengue patients with confirmed DENV-2 infection, representing approx. 70% of patients in this cohort. The DENV serotype of their first infection was unknown and we were therefore unable to stratify patients based on the serotype of their past infection”.

4a. Please add the study across HLA type to future perspective

We have added this explanation in lines 548-551: “For these analyses we utilised 15-mer overlapping peptides which capture T-cell responses restricted to any HLA type. Future studies addressing T-cell responses within specific HLA types associated with protection in dengue are required”.

4f. It would be helpful to add discussion of dysfunctional/exhaustion and the need of further studies to better address the molecular mechanism of these cells.

We have added this discussion in lines 535-537: “We report a T-cell phenotype which is consistent with T-cell dysfunction/exhaustion in SD patients, however further studies are needed to address in detail the molecular events occurring in these cells”

6. Cell-cell interaction could be included in the manuscript sup Fig. We have included these analyses in Supplementary Figure S11 (figure legend at lines 1299-1300) and a brief explanation in the results section (lines 431-434).

Reviewer #3 (Remarks to the Author):

I commend the authors for their thorough and thoughtful responses to the reviewers and the addition of new proteomic data to strengthen the conclusions. Many more details have been given with respect to data acquisition and analysis, which substantially strengthens the results.

Thank you.

I still have the following remarks:

- As secondary infection is an important risk factor, and therefore, an important confounder, I would like to see the analysis with only secondary cases included in the manuscript

Thank you for this comment. Respectively, we believe that addition of these reanalyses would not add information to the paper. We have therefore not included these in the manuscript, but we are willing to do so if the editors believe it is required to support the message of our paper. As our cohort included 118 patients with secondary infection and 5 with primary infection it is not surprising that we do not see differences in immune profiles if we exclude the 5 primary cases. While the reviewer raises an interesting point, our study was not powered to compare immune profiles in primary versus secondary infection.

- I am still not convinced one can use the terminology "progression". As the authors mention themselves, "critical phase is day 4/5-7 from onset of fever. LSA includes 11 patients: 4 were recruited prior to SD, seven classified as SD at admission. So TP1 (less

or equal than 5 days) overlaps with onset of severity (day4-5). Only when including individuals prior to day 4 after onset of symptoms), one could claim to have a prognostic immune signature.

Thank you for this comment. In the new Supplementary Fig. 2a we reanalysed our data to include only SD and non-SD patients at day 3-4 (at this time only one patient had started to display symptoms of severe dengue). These data confirm that the immune profiles we identified are already present in patients early in infection, at a time that broadly precedes the progression to severe dengue. We agree with this reviewer that further studies are needed to validate the prognostic value of our findings. This consideration has been added in lines 587-588: “Further studies validating these immune signatures as prognostic markers for SD are warranted”.

- lines 275-283: Some regions of NS3 are conserved between different DENV serotypes, so the peptide pools will contain peptides shared between DENV2 and DENV 1-4. I am not sure from these analysis how one can confirm/dismiss skewing of the immune response to the other serotypes?

We thank the reviewer for this insightful observation, and we agree with this comment- this argument led us to choose the approach we used here. Indeed, the skewing of the T-cell response in dengue was shown for few T-cell epitopes, for example for the HLA-A*1101 restricted NS3 epitope (Mongolsapaya et al Nat Med 2003). However, the T-cell response is composed of the sum of T-cell responses to each individual epitope and we need to evaluate the sum of these responses if we want to understand the impact of a skewed T-cell response to the overall response to DENV. While we cannot dismiss that T-cell skewing is occurring for some T-cell epitopes, based on our data we believe that it may not be impacting the total T-cell response to DENV. We have included these considerations in lines 542-544.

Reviewer #4 (Remarks to the Author):

I am happy that the authors have responded sufficiently to all comments.
Thank you.